# Deep-learning density functional theory Hamiltonian for efficient ab initio electronic-structure calculation

He Li [1,2,7], Zun Wang [1,7], Nianlong Zou[1,7], Meng Ye [1], Runzhang Xu[1], Xiaoxun Gong[1,3], Wenhui Duan [1,2,4,5 ✉] and Yong Xu [1,4,6 ✉]

The marriage of density functional theory (DFT) and deep-learning methods has the potential to revolutionize modern computational materials science. Here we develop a deep neural network approach to represent the DFT Hamiltonian (DeepH) of crystalline materials, aiming to bypass the computationally demanding self-consistent field iterations of DFT and substantially improve the efficiency of ab initio electronic-structure calculations. A general framework is proposed to deal with the large dimensionality and gauge (or rotation) covariance of the DFT Hamiltonian matrix by virtue of locality, and this is realized by a message-passing neural network for deep learning. High accuracy, high efficiency and good transferability of the DeepH method are generally demonstrated for various kinds of material system and physical property. The method provides a solution to the accuracy–efficiency dilemma of DFT and opens opportunities to explore large-scale material systems, as evidenced by a promising application in the study of twisted van der Waals materials.

Nowadays, ab initio calculations based on density functional theory (DFT)[1,2] have become indispensable to scientific research in physics, materials science, chemistry and biology[3], while deep learning based on neural networks has revolutionized many disciplines, from computer vision and natural language processing to scientific discoveries[4–6]. The marriage of these two important fields has led to the emerging approach of deep-learning ab initio calculations[7–24], which is contributing to the development of computational materials science. One critical problem with DFT is that it is computationally rather demanding and not very applicable to routine calculations in material systems with more than thousands of atoms. One can employ more efficient algorithms (for example, the linear-scaling methods[25]), but usually at the expense of decreasing accuracy and transferability. In principle, as a result of their expressive power, deep neural networks can learn well from DFT results and be used to bypass computationally expensive steps. The accuracy–efficiency dilemma of DFT might thus be solved by deep learning, facilitating the exploration of various important systems, including defects, disorder, interfaces, heterostructures, quasi-crystals, twisted van der Waals (vdW) materials and so on.

Tremendous efforts have been devoted to develop deep learning to find interatomic interactions or potential energies from DFT by using neural networks[7–11]. Molecular dynamics (MD) simulations in combination with deep-learning potential energies can demonstrate the efficiency of classical MD with ab initio accuracy, and the research scope of material simulation is thus greatly expanded. It is naturally desirable to generalize the deep-learning approach from the atomic simulation level to the level of electronic-structure simulation. The most fundamental quantity to be learned is the DFT Hamiltonian[26], from which almost all electron-related physical quantities in the single-particle picture can be derived, such as charge density, band structure, Berry phase and physical responses to electromagnetic fields. Instead of studying these physical quantities separately[12–17], applying the deep-learning method to the DFT Hamiltonian is an essential and challenging task. In contrast to gauge-invariant quantities, the DFT Hamiltonian matrix transforms covariantly (that is, equivariantly) under changes of the coordinate, basis and gauge, thus demanding the design of a gauge (or rotation) covariant neural network[27–29]. However, when applying a neural network to represent the relation between material structure and DFT Hamiltonian for large-scale material structures, the number of independent variables and the dimension of the Hamiltonian matrix both become infinitely large. Previous works have designed neural networks to study the DFT Hamiltonian of small molecules[18,19]. Another work considered a specific one-dimensional material and circumvented the gauge issue by learning energy eigenvalues[17]. Despite these preliminary attempts, developing a deep-learning DFT Hamiltonian to carry out electronic-structure calculations of large-scale material systems remains elusive.

In this Article we propose a general theoretical framework—the deep-learning DFT Hamiltonian (DeepH)—to study crystalline materials by means of a message-passing neural network. The challenging issues related to the (infinitely) large dimensionality and gauge (or rotation) covariance of the DFT Hamiltonian matrix are solved by virtue of locality, including the use of the local coordinate, local basis transformation and localized orbitals as basis functions. We systematically test the capability of the DeepH method by studying various representative materials with flat or curved structures, formed by strong chemical bonds or weak vdW bonds, containing single or multiple elements, excluding or including spin–orbit coupling (SOC), and so on. The example studies consistently demonstrate the high accuracy of DeepH, not only in the construction of the DFT Hamiltonian (with minor error on the scale of millielectron-volts), but also in the calculations of band- and wavefunction-related

[1]State Key Laboratory of Low Dimensional Quantum Physics and Department of Physics, Tsinghua University, Beijing, China. [2]Institute for Advanced Study, Tsinghua University, Beijing, China. [3]School of Physics, Peking University, Beijing, China. [4]Frontier Science Center for Quantum Information, Beijing, China. [5]Beijing Academy of Quantum Information Sciences, Beijing, China. [6]RIKEN Center for Emergent Matter Science (CEMS), Wako, Saitama, Japan. [7]These authors contributed equally: He Li, Zun Wang, Nianlong Zou. ✉e-mail: duanw@tsinghua.edu.cn; yongxu@mail.tsinghua.edu.cn

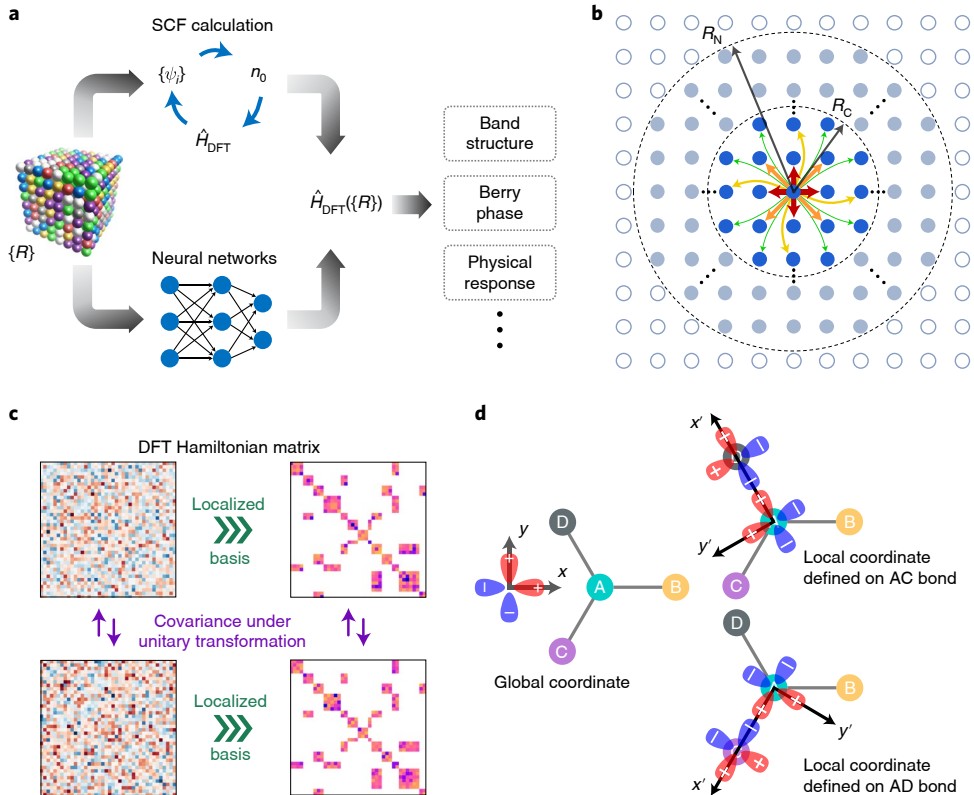

**Fig. 1 | Learning the DFT Hamiltonian $\hat{H}_{DFT}$ by virtue of locality. a**, $\hat{H}_{DFT}$ as a function of material structure (that is, atomic coordinate $\{\mathcal{R}\}$), which can be obtained by self-consistent field (SCF) calculations or learned by a neural network for efficient ab initio electronic-structure calculations. **b**, Use of the nearsightedness principle of electronic matter to learn $\hat{H}_{DFT}$, whose matrix elements in the localized basis are nonzero between neighboring atoms (within $R_C$) and influenced only by neighborhood (within $R_N$). **c**, Schematic showing the properties of the DFT Hamiltonian matrix, which is generally dense and becomes sparse in the localized basis and changes covariantly under unitary transformation. **d**, Illustration of rotation transformation for a four-atom structure with $p_{x,y}$ orbitals in varying coordinates.

physical quantities. The DeepH method performs very well in investigating twisted vdW materials in terms of accuracy, transferability and efficiency, which will be advantageous for building a twisted materials database. Our method is expected to be universal, applicable to periodic or non-periodic systems, and could find useful applications in computational materials science.

## Results

**Theoretical framework of DeepH.** One of the most fundamental problems in quantum physics is to solve the Schrödinger equation for interacting electrons of matter to predict material properties from first principles. The use of DFT[1,2] has been recognized for this purpose, and it replaces the complicated many-body problem with a simpler auxiliary one, $\hat{H}_{DFT}|\psi\rangle = \mathcal{E}|\psi\rangle$, describing non-interacting electrons with interacting density[30], where $\hat{H}_{DFT}$ is the DFT Hamiltonian operator, and $\mathcal{E}$ and $|\psi\rangle$ are the Kohn–Sham eigenvalue and eigenstate, respectively. Typically, the ab initio DFT Hamiltonian $\hat{H}_{DFT}$ is obtained via self-consistent field calculations, followed by calculations of material properties (Fig. 1a). According to the Hohenberg–Kohn theorem[1], there is a one-to-one correspondence between the external field determined by the material structure $\{\mathcal{R}\}$ and $\hat{H}_{DFT}$, implying a mapping function: $\{\mathcal{R}\} \mapsto \hat{H}_{DFT}(\{\mathcal{R}\})$. The generic form of $\hat{H}_{DFT}(\{\mathcal{R}\})$, however, is too complicated to be expressed analytically, but can be represented by the DeepH method. For generally non-periodic crystalline materials containing an infinite number of atoms, $\hat{H}_{DFT}(\{\mathcal{R}\})$ has an infinite number of independent variables in $\{\mathcal{R}\}$. Therefore, the DFT Hamiltonian matrix may have an infinitely large dimension

and the matrix is invariant under atom permutation and translation, and covariant under rotation and gauge transformations (Fig. 1b). In this sense, learning the DFT Hamiltonian is much more challenging than learning scalar physical quantities, such as total energy[7–10].

Next we show that the problem of learning the DFT Hamiltonian, although appearing formidable, can be solved by virtue of the locality. As revealed by Kohn and colleagues, local physical properties do not respond to distant changes of external potential due to the destructive interference between the many-particle eigenstates[31,32]. This implies a widely applicable principle of locality or 'nearsightedness' in electronic matter. Thus, there is no need to study the entire system at once, and only information of the neighborhood is relevant for learning the DFT Hamiltonian (Fig. 1b).

A proper selection of basis sets is essential to DeepH. DFT calculations usually use plane waves or localized orbitals as basis functions. The latter is compatible with the locality and non-periodicity nature of the problem and thus will be employed. $\hat{H}_{DFT}$ is then expressed as a sparse matrix (Fig. 1c) benefiting from the local or semilocal property of the Kohn–Sham potential. The matrix element $H_{i\alpha,j\beta}$ ($\alpha$ and $\beta$ refer to localized orbitals centered at atoms $i$ and $j$) vanishes when the distance between atoms $i$ and $j$ is larger than a cutoff radius $R_C$. $R_C$ is determined by the spread of localized orbitals, which is on the order of ångstroms, much smaller than the nearsightedness length $R_N$ (Fig. 1b). We suggest using non-orthogonal atomic-like orbitals. These are typically more localized than orthogonal ones as a result of circumventing the conflicting requirements of localization and orthogonalization[33]. Moreover, their gauge is system-independent, and the rotation transformation is well described

by spherical harmonics. In contrast, the widely used Wannier functions do not possess such advantages[34]. By taking advantage of the sparseness and nearsightedness, only Hamiltonian matrix blocks $H_{ij}$ between neighboring atoms (within $R_C$) have to be learned, and only information about the neighborhood of atoms $i$ and $j$ (within $R_N$) is relevant to learning $H_{ij}$.

A critical issue is to deal with covariant transformations of the DFT Hamiltonian matrix. The Hamiltonian matrix itself is not physically observable, and changes covariantly when varying coordinate, basis or gauge globally or locally. Taking a four-atom structure as an example (Fig. 1d), a global rotation of the atomic structure (or basis functions) changes the DFT Hamiltonian matrix. The new Hamiltonian matrix is related to the original one by a rotation transformation. The local transformation is less obvious. In this example structure, the atom pairs AB, AC and AD share the same local chemical environment, and the Hamiltonian matrix blocks are related to each other by rotation transformations. Specifically, the transformed Hamiltonian matrix block $H'_{AC}$ ($H'_{AD}$) coincides with $H_{AB}$ under a clockwise rotation of the basis functions by 120° (240°) for AC (AD). In infinite crystalline materials, we may encounter atom pairs with varying orientations. It is thus difficult (if not impossible) to learn the covariant relations by neural network via data augmentation[18]. In this Article we propose a strategy to help DeepH work efficiently and accurately via local coordinates (details are described in Supplementary Section 3; see Supplementary Information for details of the computational methods and results, which includes refs. [18,19,26]), in which the locally transformed Hamiltonian matrix blocks $H'_{ij}$ are invariant under rotation (Fig. 1d). By changing the coordinate from local back to global, a rotation (or basis) transformation is applied to $H'_{ij}$. The obtained $H_{ij}$ will then naturally satisfy the covariant requirement.

**Neural network architecture of DeepH.** Next we present a deep neural network representation of the DFT Hamiltonian based on a message-passing neural network (MPNN)[35], which is widely applied in materials studies[8,10,36–38]. The rules of constructing crystal graphs and the MPNN are illustrated in Fig. 2a. Each atom is represented by a vertex, and atom pairs (with a distance smaller than $R_C$) are represented by edges. The MPNN uses edge embeddings to represent $H'_{ij}$. Self-loop edges are added in the graph to account for intra-site couplings. Let $v_i$ and $e_{ij}$ denote the vertex feature of atom $i$ and the edge feature of atom pair $ij$, respectively. The initial vertex features are the embeddings of atomic number $Z_i$, and the initial edge features are the interatomic distance $|r_{ij}|$ expanded with the Gaussian basis, centered at different points $r_n$:

$$v_i^{(0)} = \text{Embedding}(Z_i), \tag{1}$$

$$e_{ij}^{(0)} = \exp(-(|r_{ij}| - r_n)^2/\sigma^2). \tag{2}$$

The architecture and workflow of the MPNN are presented in Fig. 2b. In a message-passing (MP) layer, the vertex and edge features are updated successively as follows:

$$v_i^{(l)} = \text{LayerNorm}\left(\sum_{k \in \mathcal{N}_i} \Phi_v^{(l)}\left(z_{ik}^{(l-1)}\right)\right) + v_i^{(l-1)}, \tag{3}$$

$$e_{ij}^{(l)} = \Phi_e^{(l)}\left(v_i^{(l)} \| v_j^{(l)} \| e_{ij}^{(l-1)}\right), \tag{4}$$

where $\mathcal{N}_i$ is a set containing neighboring vertices with edge connection to vertex $i$, $\|$ denotes the concatenation of feature vectors, superscript $l$ refers to the $l$th MP layer, $z_{ik}^{(l-1)} \equiv v_i^{(l-1)} \| v_k^{(l-1)} \| e_{ik}^{(l-1)}$ is the concatenation of vertex and edge features of the neighborhood,

layer normalization[39] is employed to improve training efficiency, and $\Phi_v^{(l)}$ and $\Phi_e^{(l)}$ are neural networks for updating vertex and edge features, respectively. The local chemical environment of the neighborhood within $R_C$ will be aggregated in an MP layer. As MP layers are stacked, more and more information of the distant chemical environment will be aggregated into the features, enabling the learning of $H'_{ij}(\{\mathcal{R}\}_N)$.

A problem about the local coordinate should be noted. Because the local coordinate is defined for each edge according to its local chemical environment, sometimes minor modifications of local structures could substantially change the coordinate axes, making the transformed $H'_{ij}$ considerably different and thus leading to inaccuracy in the deep learning. We find that the problem is solvable by introducing one local coordinate message-passing (LCMP) layer after several MP layers. In the LCMP layer, orientation information (unit vector $\hat{\mathbf{r}}_{ik}^{pq}$) of bond $ik$ relative to the local coordinate defined for edge $pq$ is added into the initial edge features, where $i$, $k$, $q$ and $p$ are all atomic indices. $\theta_{ik}^{pq}$ and $\phi_{ik}^{pq}$ are the corresponding polar and azimuthal angles, respectively, of the reference atoms $i$, $k$, $q$ and $p$. The orientation information based on bonds between the central atom and its neighbors was introduced for the study of total energy in ref. [9]. The vertex and edge features ($v_i^{pq(L)}$ and $e_{ij}^{pq(L)}$) defined for local coordinate $pq$ are updated as follows:

$$v_i^{pq(L)} = \sum_{k \in \mathcal{N}_i} \Phi_v^{(L)}\left(z_{ik}^{(L-1)} \| \{Y_{Jm}(\theta_{ik}^{pq}, \phi_{ik}^{pq})\}\right), \tag{5}$$

$$e_{ij}^{pq(L)} = \Phi_e^{(L)}\left(v_i^{pq(L)} \| v_j^{pq(L)} \| e_{ij}^{(L-1)}\right), \tag{6}$$

where a set of real spherical harmonic functions $\{Y_{Jm}\}$ are used to capture orientation information (where $J$ is an integer ranging from 0 to 4 and $m$ is an integer between $-J$ and $J$.), and $e_{ij}^{ij(L)}$ will be used to represent $H'_{ij}$. Note that the introduction of the LCMP layer into DeepH is critical to improving the prediction accuracy according to our test (Supplementary Table 3). Finally, $H_{ij}$ is calculated from $H'_{ij}$ via rotation transformation. The neural network of DeepH is trained by DFT data and then applied to predict the DFT Hamiltonian for unseen atomic structures, which can bypass the time-consuming DFT self-consistent calculation and enable efficient electronic-structure calculations.

**Capability of DeepH.** Example studies were performed on various representative materials, including graphene, $MoS_2$ and their curved counterparts (that is, nanotubes), as well as moiré-twisted materials with negligible or strong SOC. The DFT Hamiltonian was computed by using 13, 19, 13 and 19 non-orthogonal atomic-like basis functions for C, Mo, S and Bi, respectively. The MPNN model, including five MP layers followed by one LCMP layer, was trained by minimizing the loss function defined as the mean-squared errors of $H'_{i\alpha, j\beta}$. Once $\hat{H}_{DFT}(\{\mathcal{R}\})$ is learned by the neural network, various kinds of physical property, such as band structure, Berry phase and physical responses to external fields, can be predicted while bypassing DFT self-consistent calculations (Fig. 1a). To check the reliability of our method, we studied eigen-energy-based quantities (density of states (DOS) or bands) as well as wavefunction-related properties (optical transition and shift current). Shift current is of particular interest, being an important photovoltaic effect generated by nonlinear optical progress and closely related to topological quantities (for example, Berry phase and curvature)[33,40,41]. The linear and nonlinear optical responses were studied using methods developed by us[33,41].

The training of a neural network generally demands a large amount of data. In our study, 5,000 configurations of the graphene 6×6 supercell were generated by ab initio MD at a temperature of

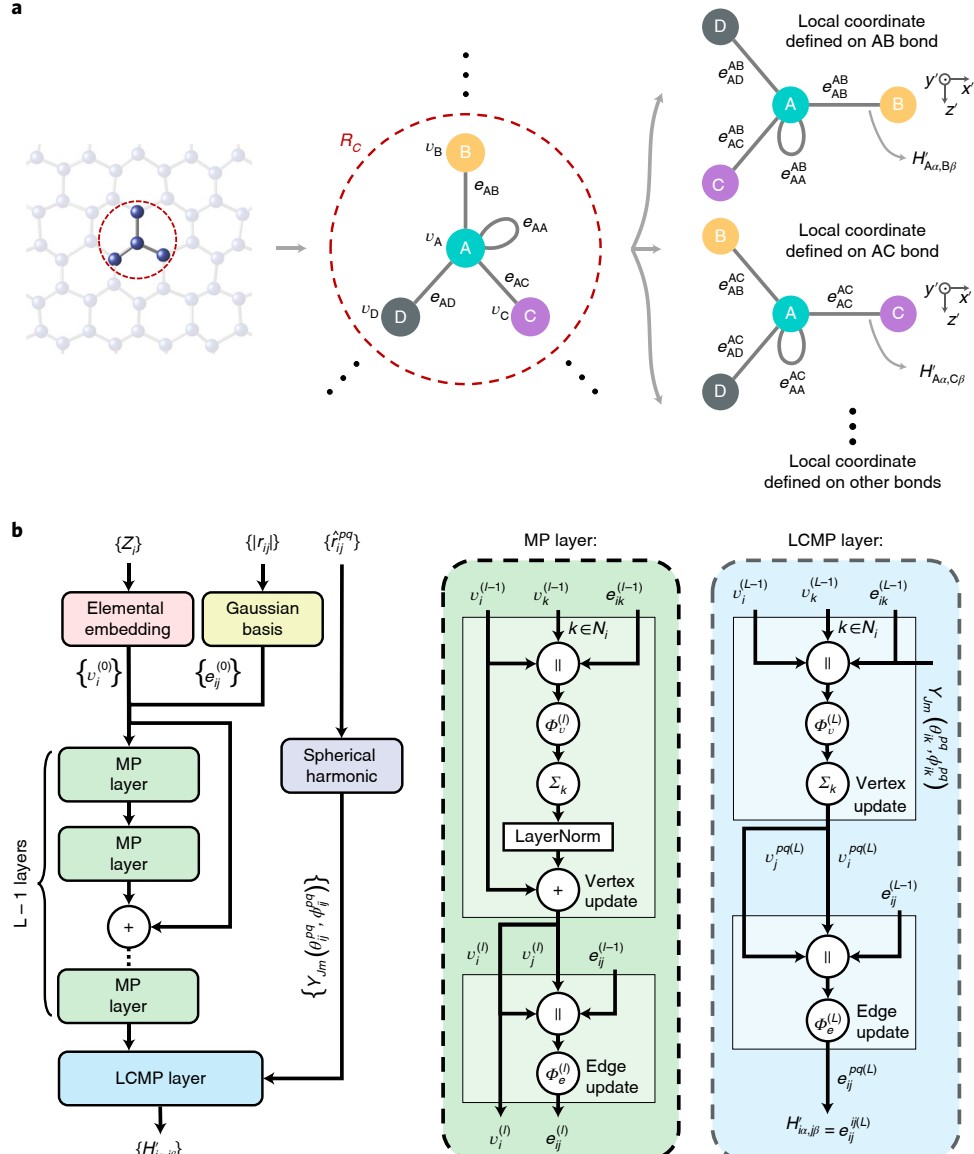

**Fig. 2 | Crystal graph and MPNN including *L* layers employed by DeepH. a**, Crystal graph with vertices $v_i$ and edges $e_{ij}$ used for the MPNN. For simplicity, only edges connected to the nearest neighbors are shown. $R_C$ denotes the cutoff radius. In the LCMP layer, different crystal graphs with new edges, $e_{ij}^{pq}$, are applied for different local coordinates defined on varying atom pairs *pq*. **b**, Architecture and workflow of the deep neural network, including $L-1$ MP layers with atomic numbers $\{Z\}$ and interatomic distances $\{|r_{ij}|\}$ as initial inputs and one LCMP layer using additional orientation information $\{\hat{r}_{ik}^{pq}\}$ relative to different local coordinates.

300 K, giving 14,400,000 nonzero Hamiltonian matrix blocks. A total of 270 configurations were used for training, which is large enough to ensure convergence, as demonstrated by the calculated learning curve as a function of training set size (Supplementary Fig. 4), 90 configurations were used for hyperparameter optimization, and the remainder for the test. The mean absolute error (MAE) of $H'_{i\alpha,j\beta}$ for the test set is shown in Fig. 3a. The MAE value averaged over all $13 \times 13$ orbital combinations was 2.1 meV, and the individual values were distributed between 0.4 meV and 8.5 meV. This MAE is quite small considering that the Hamiltonian matrix element is typically on the order of electronvolts. For example, $H'_{i\alpha,j\beta}$ for the 1*s* orbital and the nearest neighbor obtained from DFT calculations has a mean value of −10.1 eV and a standard deviation (s.d.) of 315 meV (Fig. 3b), whereas the corresponding MAE of DeepH is 6.6 meV, corresponding to a high coefficient of determination, $r^2 = 0.9994$. For another 2,000 unseen configurations of a graphene supercell

sampled by ab initio MD from 100 K to 400 K, the generalization MAE of $H'_{i\alpha,j\beta}$ was as small as 1.9 meV on average, demonstrating the high accuracy of DeepH.

Figure 3c,d shows the results for the DOS and shift current conductivity, respectively. For the 2,000 unseen configurations of graphene, the MAE between the predicted and calculated DOS for 500 points between −6 eV and +6 eV around the Fermi level is on the order of $0.1 \times 10^{-3}\,\mathrm{eV^{-1}\,\mathring{A}^{-2}}$, much smaller than the absolute values (usually $>10 \times 10^{-3}\,\mathrm{eV^{-1}\,\mathring{A}^{-2}}$). The spectra for the DOS and shift conductivity were compared for DeepH and DFT and show satisfactory agreement.

DeepH uses the embedding of atomic numbers as initial vertex features and can naturally work for systems containing multiple atomic types. For demonstration, we performed calculations on monolayer $MoS_2$, used 300 random $5 \times 5$ $MoS_2$ supercell structures for training, and achieved high accuracy (Fig. 4b–d). Specifically,

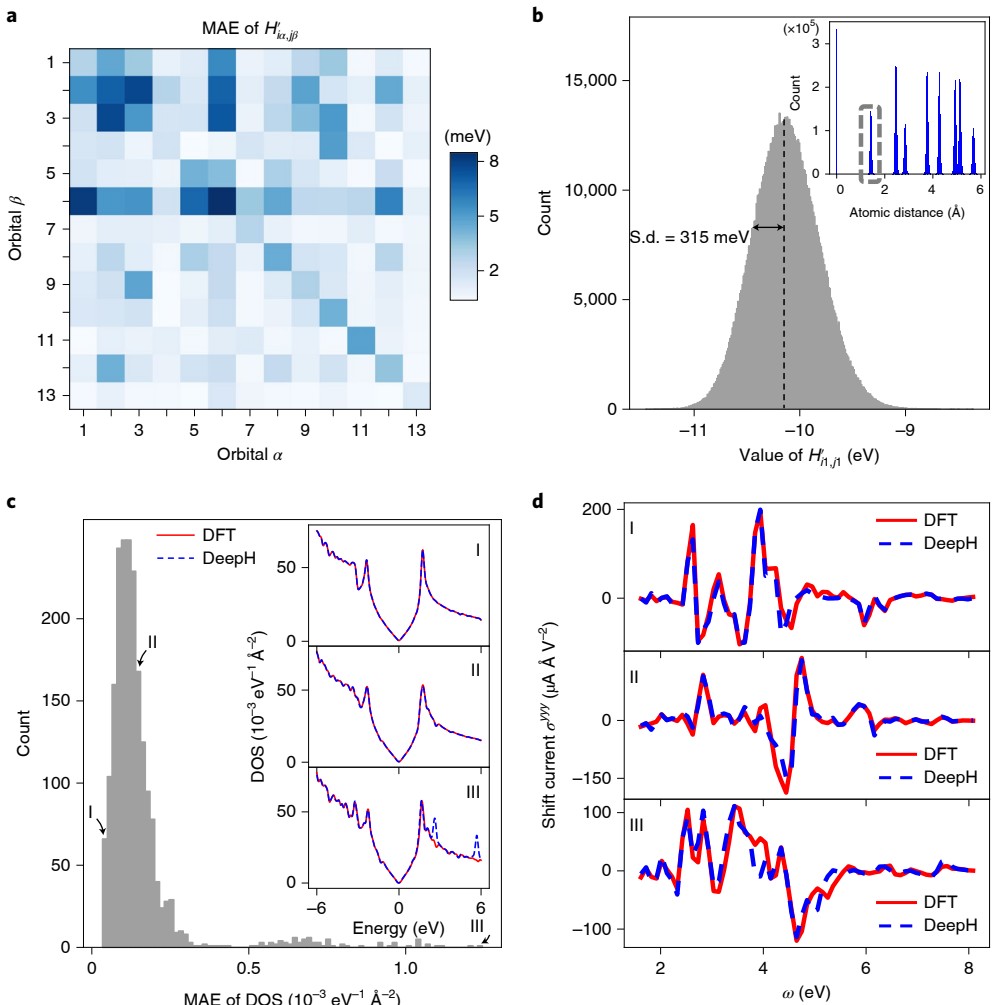

**Fig. 3 | Performance of DeepH on studying graphene. a**, MAE of $H'_{i\alpha,j\beta}$ for different orbitals. **b**, Distribution of $H'_{i1,j1}$ for the nearest neighbors (atomic distance between 1.28 and 1.6 Å; see the corresponding distribution in the inset). The s.d. of the computed $H'_{i1,j1}$ is 315 meV for the test set. **c,d**, Distribution of the generalization MAE of the DOS for 2,000 unseen material structures (**c**). Three typical structures with the best, median and worst MAE values for the DOS (atomic structures included in Supplementary Data 1) are indicated. Their DOS (**c**, inset) and shift current conductivity $\sigma^{yyy}$ (**d**), computed by DFT and DeepH, are compared.

the averaged MAE of $H'_{i\alpha,j\beta}$ for Mo–Mo, Mo–S, S–Mo and S–S atom pairs are as low as 1.3, 1.0, 0.7 and 0.8 meV, respectively. The predicted material properties (band structure, electric susceptibility and shift current conductivity) match well with DFT self-consistent calculations (Fig. 4 and Supplementary Figs. 6–8). The results indicate that DeepH works well for systems containing multiple atomic types, and at no obvious expense of increased computational complexity.

We also tested the generalization ability of DeepH by making predictions on new structures that were unseen in the training set (Fig. 5a). Test samples of carbon nanotubes (CNTs) and $MoS_2$ nanotubes were selected for this purpose, as these have a curved geometry that is suitable for checking the rotation covariance of the method. For CNTs, the averaged MAE of the DFT Hamiltonian matrix is insensitive to nanotube chirality and monotonically decreases with increasing nanotube diameter $d$, reducing to below 3.5 meV for $d > 2$ nm (Supplementary Fig. 5a). For a zigzag (25,0) CNT ($d \approx 2$ nm), the predicted band structure (Fig. 5b) and other physical properties (such as electric susceptibility; Supplementary Fig. 5b) reproduce the DFT calculation results well. Similar results were obtained for a large-diameter $MoS_2$ nanotube (Fig. 5c). Note that it is computationally very expensive to study large-diameter

nanotubes with DFT. In contrast, their physical properties can be accurately predicted by DeepH at much lower computational expense.

Next we compared the computational cost of DFT and DeepH in constructing DFT Hamiltonian matrices for flat supercells and curved nanotubes of graphene and $MoS_2$ (Supplementary Table 1). Compared to DFT (for which the computational time grows approximately cubically with system size), the computational time of DeepH grows linearly with system size and the prefactor is much smaller. For the example study of a $MoS_2$ $35 \times 35$ supercell, DeepH reduces the computation time by three orders of magnitude. This improvement would become even more considerable with increasing system size. We have thus demonstrated the high efficiency of DeepH in dealing with large-scale material systems.

**Application to twisted vdW materials.** Twisted bilayer graphene (TBG)—or twisted vdW materials in general—are rising stars of materials science, with the 'magic' moiré twist providing opportunities to explore exotic quantum phases, such as the correlated insulator, unconventional superconductivity, the (fractional) Chern insulator and so on[42–46]. Despite their enormous impact, investigating the twist-angle dependence of material properties

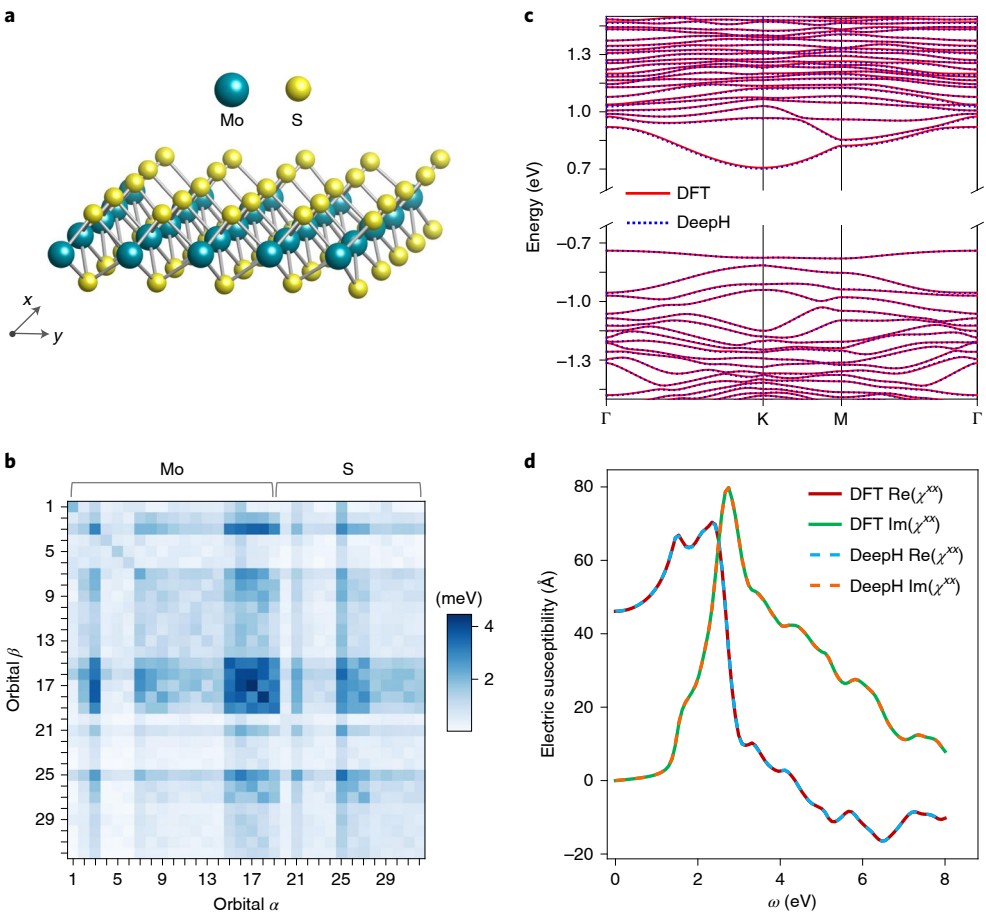

**Fig. 4 | Performance of DeepH on studying monolayer MoS₂. a**, Atomic structure of $MoS_2$. **b**, MAE of $H'_{i\alpha,j\beta}$ for different orbitals. **c,d**, Band structures (**c**) and real and imaginary parts (**d**) of electric susceptibility $\chi^{xx}$ computed by DFT and DeepH for a 5×5 $MoS_2$ supercell. A representative structure with the median generalization MAE of the DFT Hamiltonian matrix (the atomic structure is presented in Supplementary Data 2) is considered in **c** and **d**. Γ, K and M represent different high-symmetry *k*-points of the Brillouin zone (the same applies for Figs. 5b,c and 6b,c).

remains a great challenge, both experimentally and theoretically. Theoretically, empirical tight-binding and continuum models work well for simple model systems of TBG[42], but are usually not accurate enough to study other materials. Indeed, ab initio calculations need to accurately describe the electronic structure, but are only applicable to small moiré supercells. In short, the theoretical study of twisted vdW materials is limited by the accuracy–efficiency dilemma[47]. DeepH is designed to solve the dilemma, and works well for studying twisted materials, as we will show.

The workflow for using DeepH to study twisted materials is displayed in Fig. 6a. First, the training data are obtained by DFT calculations of non-twisted structures, which usually contain hundreds of randomly perturbed samples of a relatively small supercell. The process of generating datasets is largely simplified because there is no need to consider varying twist angles for training. Second, the neural network of DeepH is trained using the DFT data. Finally, the trained DeepH is applied to predict the DFT Hamiltonian and calculate the material properties for new twisted structures with an arbitrary twist angle θ.

As a proof of principle, we first considered TBGs that have already been intensively studied[42–46]. The neural network of DeepH, once trained by DFT data for zero twist angle, is able to give highly accurate predictions on material properties for varying twist angles. The good transferability of DeepH was demonstrated by comparing its results with those calculated with DFT. The averaged MAE of $H'_{i\alpha,j\beta}$ is as low as sub-millielectronvolt when testing moiré-twisted

supercells of up to ~1,000 atoms (Supplementary Fig. 9). Because of the high accuracy in predicting the DFT Hamiltonian, the calculated band structures from DeepH match the DFT results well (Fig. 6b and Supplementary Fig. 10), and similar agreements are thus expected for other material properties. By using traditional DFT methods it is difficult to study TBG with magic angle $\theta \approx 1.08°$ and including 11,164 atoms per supercell, but this can be achieved quite easily with DeepH. For this large-sized structure, the uncertainty of an ensemble of neural networks can serve as a reliability indicator of accuracy[48]. The corresponding results indicate that the high prediction accuracy is preserved for magic-angle TBG (Supplementary Fig. 11). Indeed, the band structure calculated by DeepH satisfactorily matches the DFT benchmark result[49] that was obtained by using the plane-wave basis at enormous computational cost (Fig. 6b, right). Importantly the existence of flat bands near the Fermi level, a characteristic feature of the magic angle, is well reproduced by DeepH.

Our method works well not only for TBGs, but also for other twisted vdW materials. Special attention has been paid to materials with strong SOC, such as twisted bilayer bismuthenes (TBBs), where the interplay between the strong SOC and moiré twist induces exotic physical properties[50,51]. In contrast to the Hamiltonian matrix calculated without SOC, the DFT Hamiltonian matrix with SOC has complex values and needs to take the spin degree of freedom into account for rotation transformation. Despite the additional complexity, a high prediction accuracy comparable to that of

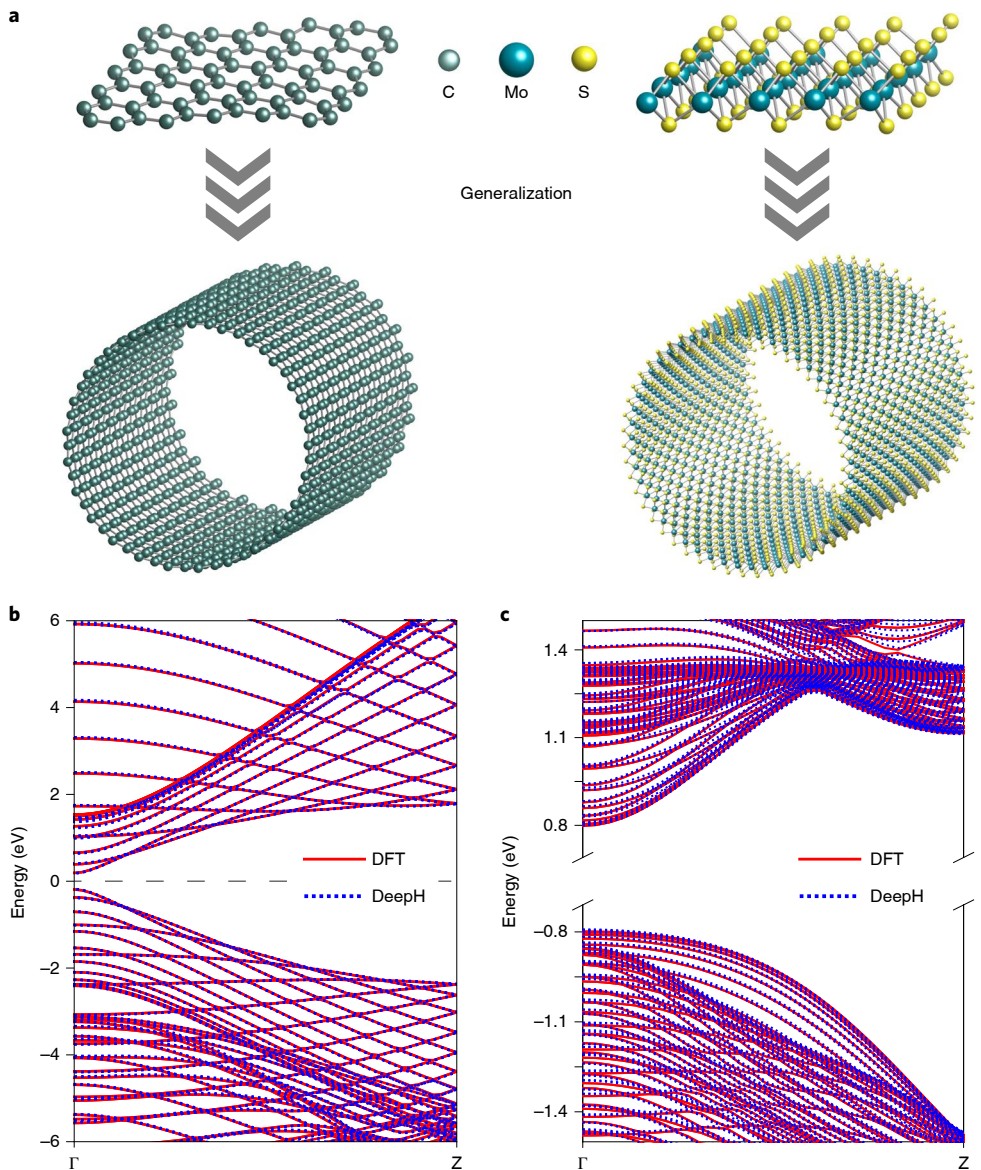

**Fig. 5 | Generalization ability of DeepH, from flat sheets to curved nanotubes. a**, The use of DeepH, trained by DFT results for flat sheets, to study curved nanotubes: graphene (left) and $MoS_2$ (right). **b,c**, Band structures for a zigzag $(25, 0)$ carbon nanotube (**b**) and a zigzag $(50, 0)$ $MoS_2$ nanotube (**c**), computed by DFT and DeepH. The Fermi level is aligned at the middle of the bandgap.

TBGs is achieved for TBBs on predicting the DFT Hamiltonian (Supplementary Fig. 13) as well as on calculating material properties (Fig. 6c and Supplementary Fig. 14).

It is worth noting that the computational time can be reduced considerably by replacing DFT self-consistent field iterations with DeepH, making ab initio electronic-structure calculations much more efficient and applicable to much larger material systems (Fig. 6d), such as magic-angle TBG. On the other hand, compared to empirical tight-binding and continuum models, DeepH has slightly lower efficiency, but it has much better accuracy and transferability. Moreover, superior to empirical methods, DeepH can easily and appropriately treat SOC, which is advantageous for exploring spin-related or topological quantum phenomena. For comparison, the performance of different theoretical methods is summarized in Supplementary Table 5. DeepH can outperform the currently used approaches in studying twisted materials, and the method is promising for studying twist-angle-dependent physical properties and for building twisted materials databases.

**Wide applicability of DeepH.** Many types of deep-learning DFT method have been developed so far[7–10,12–24]. They can be classified into two groups that aim to improve either the accuracy or efficiency of DFT via deep-learning techniques. Representative works of the first group have achieved substantial breakthroughs recently in developing advanced exchange and correlation functionals via deep neural networks[20–22]. The second group of works try to reproduce DFT results via deep learning, in a similar manner to DeepH. Among these, great successes have been achieved regarding deep-learning potential[7–10], facilitating highly efficient ab initio atomic-structure calculations. However, the corresponding developments of ab initio electronic-structure methods are at a preliminary stage. Most current works select a one-step strategy and directly learn individual physical quantities, such as bandgap, band dispersion, electron density and wavefunction[12–17]. The two-step strategy employed by DeepH, which first learns the DFT Hamiltonian and then predicts the desired physical properties, is advantageous in two regards. First, all the above-mentioned electron-related physical

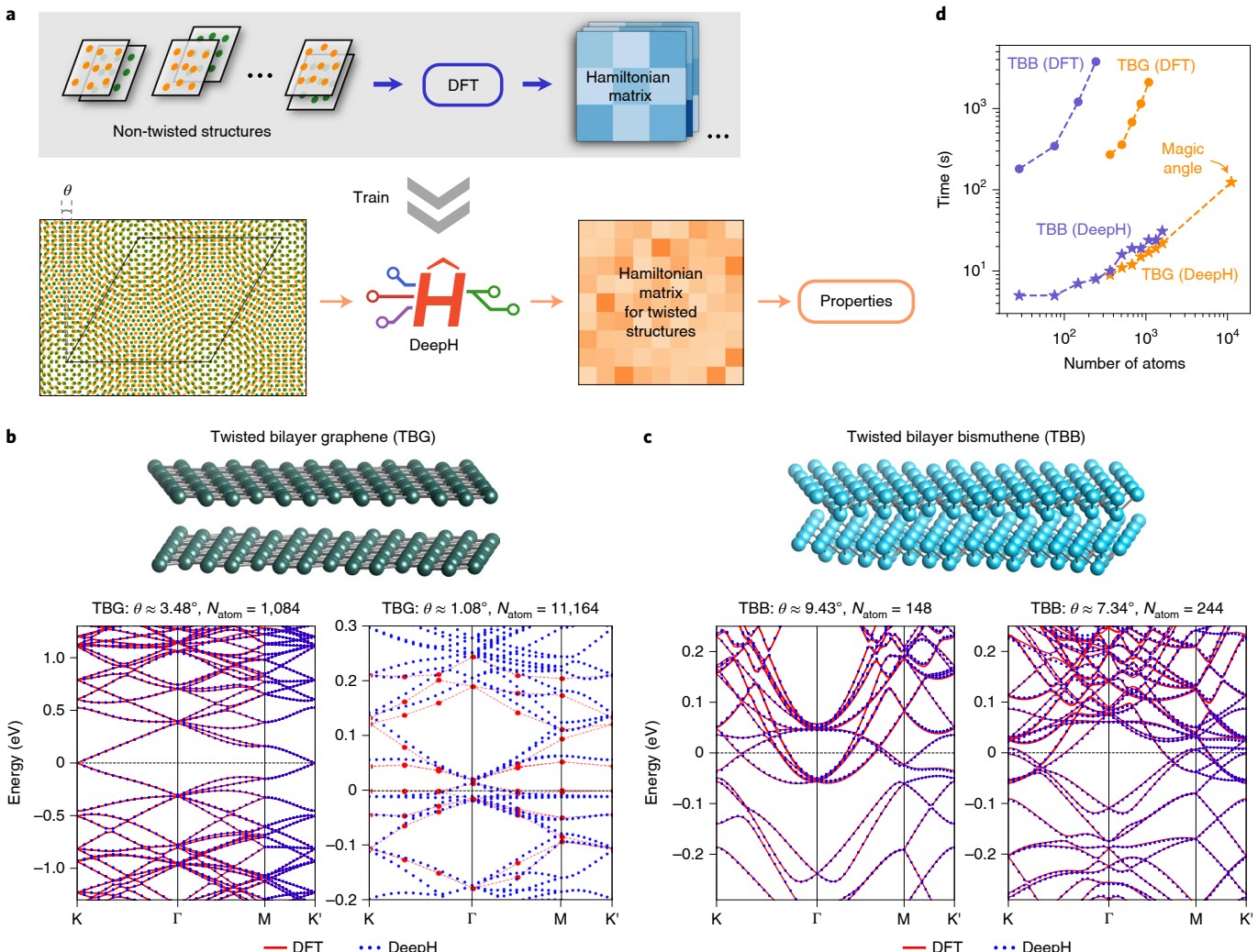

**Fig. 6 | Application of the DeepH method to study moiré-twisted materials. a,** Workflow for studying twisted materials using DeepH, which uses the DFT results for small non-twisted structures as training data and makes predictions on twisted structures with arbitrary twist angle θ. **b,c,** Band structures computed by DFT and DeepH for TBGs (**b**: θ ≈ 3.48°, 1.08°) and TBBs (**c**: θ ≈ 9.43°, 7.34°). In **c**, the DFT bands for the magic angle θ ≈ 1.08° are adapted from ref. [49]. **d,** Computation time to construct the DFT Hamiltonian matrices of TBGs and TBBs with varying system size by DFT self-consistent calculations versus by DeepH. For comparison, the calculations were all done by one compute node equipped with two AMD EPYC 7542 central processing units (CPUs), although DeepH works much more efficiently on graphics processor unit (GPU) nodes.

quantities can be simultaneously derived from DeepH. Second, and more importantly, the complex structure–property relation can be accurately described by DeepH, as we have demonstrated here, by benefiting from the nearsighted nature of the DFT Hamiltonian. In contrast, the nearsightedness principle is not applicable to some physical quantities, such as band structure and wavefunction.

Recently, Hegde and Bowen attempted to study the DFT Hamiltonian via statistical learning (not deep learning)[26], which works for small unit cells of simple metal copper. However, this method can hardly be applied to study more complex material systems due to the limited expressive power of statistical learning and the lack of an appropriate treatment of rotation covariance. A quantitative comparison between this method and DeepH is presented in Supplementary Section 6. A few primary deep-learning works have been applied to small molecules[18,19] and are applicable to systems with a fixed number of atoms. Distinct from existing deep-learning DFT methods, the DeepH method shows excellent performance on studying periodic or non-periodic crystalline materials in terms of accuracy, efficiency and transferability, as demonstrated by case studies on various quasi-one-dimensional (quasi-1D) and

2D materials without or with multiple elements, curved geometry or moiré twist. DeepH can also be applied to study material systems of other space dimensions. For example, we have carried out experiments on 3D bulk materials (including silicon and allotropes of carbon) as well as quasi-0D molecules (Supplementary Section 6). With the help of DeepH, the accuracy–efficiency dilemma of DFT can be solved and efficient ab initio electronic-structure calculations become feasible for large-scale material systems.

One may straightforwardly check the generalization ability of DeepH by performing principal component analysis (PCA) for the output atom features of the final MP layer or the output bond features of the final LCMP layer. We performed PCA on monolayer sheets versus nanotubes and non-twisted versus twisted bilayers, as presented in Supplementary Figs. 17–20. The corresponding results are discussed in Supplementary Section 5. It was established from the PCA results that DeepH can make accurate predictions on new structures with principal components substantially different from the original training set, showing satisfactory generalization ability.

It is worthwhile comparing our method with covariant neural network methods (such as tensor-field networks[27], Cormorant[28],

PhiSNet[19] and so on), which are based on spherical harmonic functions and group representation theory. These methods require tensor products that use Clebsch–Gordan coefficients in every layer of the neural network during training and inference processes to ensure rotational covariance. The tensor-product computation could be very expensive, especially for large-sized systems and for calculations involving basis sets of high orbital angular momenta. As far as we know, applying these methods to study the electronic structure of large-scale material systems remains elusive.

In contrast, our method only needs to perform the basis transformation once before the training process, which is computationally very efficient. Moreover, benefitting from the rotation-invariant nature of the local coordinates, our approach can apply a rotation-invariant neural network to predict rotation-covariant quantities, making the neural network architecture more flexible and efficient. Importantly, further development of the method would benefit from the great developments in transformation-invariant neural networks. Because all the important local bonding information, including bond length and orientation information, has been included as input, our method is expressive enough to achieve high prediction accuracy. Quantitative comparisons against the tensor-product-based method on studying molecule datasets[18,19] indicate that DeepH can achieve comparable accuracy with much less computation time and a smaller number of parameters (Supplementary Table 4).

Deep neural networks, in principle, can be applied to deal with complex problems with a large configuration space due to their expressive power. The object of the present work is to learn the DFT Hamiltonian matrix as a function of atomic positions. For most physical problems, only atomic configurations near equilibrium positions are of concern thermodynamically due to their relatively low energies. We thus focused on the configuration space near equilibrium for a given material. Solids with nearly periodic structures (like graphene with lattice vibrations) usually have a small configuration space. The DeepH method can work well for such kinds of system. On the other hand, training a model in a large configuration space is much more challenging and usually requires more training data and possibly demands methodological improvements to achieve good accuracy.

To test the performance of DeepH, we considered two kinds of material system with a relatively larger configuration space: (1) 3D bulk structures including different allotropes of carbon (graphite and diamond) and (2) quasi-0D molecules. For the former, one unified neural network was applied to predict the DFT Hamiltonian for the two kinds of carbon allotrope. The MAEs of DeepH do not increase with respect to that of graphene, although the configuration space becomes larger. For the latter, we studied molecules of increasingly larger size (from 3 to 21 atoms) to consider the growth of the configuration space. Their averaged MAEs of $H'_{i\alpha,j\beta}$ are on the sub-millielectronvolt order, also lower than that for graphene. More detailed results are presented in Supplementary Sections 4 and 6. These experiments suggest that DeepH is very probably applicable to the study of material systems spanning a large configuration space. We would like to do more critical experiments and developments in future works.

## Discussion

We have proposed a general framework to represent the DFT Hamiltonian by a deep neural network, which universally builds a mapping from materials structures to physical properties. The method extends the scope of first-principles research and opens opportunities to investigate fundamental physics and large-scale material systems, such as twisted vdW materials. However, the current DeepH framework is not without limitations. For example, the trained model has only been applied to study unseen materials that have a chemical bonding environment close to the dataset.

To investigate material systems with a strongly varying chemical environment, one still needs to manually design an appropriate dataset to improve the training efficiency. Automatic construction of the dataset and on-the-fly optimization of the training process could be explored in the future.

Some generalizations of the method are straightforward, whereas some others are not. For example, the method can be generalized to study large-scale systems without periodicity (for example, non-commensurate twisted materials or quasi-crystals). Some other material systems (for example, disorder, defects[23] and interfaces) in principle can be described by DeepH as well, but demand more training data to learn the varying chemical environment. Moreover, DeepH is compatible with DFT not only for exchange correlation functionals in the local density approximation or generalized gradient approximation (GGA), but also for the more advanced functionals, such as meta-GGA, hybrid functionals and so on. Note that the hybrid functionals demand a larger cutoff radius for constructing crystal graphs than usual. Furthermore, by combining deep-learning potential and DeepH together, efficient MD simulation and electronic-structure calculations can be performed simultaneously, making the real-time simulation of electron–lattice coupling possible. Another valuable extension of this current work is the combination of DeepH and efficient linear algebra algorithms (for example, diagonalization for large sparse matrices and linear algebra algorithms on GPUs), which could further improve the computational efficiency and promote exploration of mesoscopic physics and materials. There is much room for future development of the method, which we would like to consider in future works.

## Methods

**Dataset preparation.** We generated random structural configurations of $6 \times 6$ monolayer graphene and $5 \times 5$ monolayer $MoS_2$ supercells by ab initio MD calculations using the Vienna ab initio simulation package[52]. Simulations were performed with the projector-augmented wave[53,54] pseudopotentials and the GGA parameterized by Perdew, Berke and Ernzerhof (PBE)[55]. The cutoff energy of the plane waves was 450 eV and only the Γ point was used in our k-mesh. For the monolayer graphene supercells, two simulations were carried out under the canonical ensemble: one with a constant temperature of 300 K and the other with temperature increasing from 100 K to 400 K. With a time step of 1 fs, our dataset consisted of 5,000 frames obtained at 300 K and 2,000 frames obtained between 100 K and 400 K. As for the monolayer $MoS_2$ supercells, 1,000 random atomic structures of $5 \times 5$ supercells were generated by ab initio MD calculations performed at 300 K with a time step of 1 fs.

Furthermore, to train DeepH models for moiré-twisted vdW materials, we prepared datasets for TBGs (TBBs) from zero-twist-angle $4 \times 4$ ($3 \times 3$) bilayer supercells by shifting one of the two vdW layers within the 2D plane and subsequently inserting random perturbations to each atomic site. The interlayer spacing of the fully relaxed bilayer unit cells with the most energetically favorable stacking was used to construct the training dataset and moiré-twisted supercells (3.35 Å for TBG and 3.20 Å for TBB). In total, 300 and 576 shifted and perturbed supercell structures were included in the datasets for TBG and TBB, respectively.

We calculated DFT Hamiltonians with pseudo-atomic localized basis functions as implemented in the OpenMX software package version 3.9[56,57]. Calculations were performed with the PBE exchange correlation functional and norm-conserving pseudopotentials[58]. For monolayer graphene, CNTs and TBGs, C6.0-s2p2d1 pseudo-atomic orbitals were used, including 13 atomic-like basis functions, with a cutoff radius of $R_C = 6.0$ Bohr. For monolayer $MoS_2$ and $MoS_2$ nanotubes, Mo7.0-s3p2d2 and S7.0-s2p2d1 pseudo-atomic orbitals were used, including 19 atomic-like basis functions for Mo and 13 for S ($R_C = 7.0$ Bohr). For TBBs, Bi8.0-s3p2d2 pseudo-atomic orbitals were used, including 19 atomic-like basis functions ($R_C = 8.0$ Bohr). The energy cutoff was set to 300 Ry. A Monkhorst–Pack k-mesh of $5 \times 5 \times 1$ was used for supercells of monolayer graphene with 72 atoms, monolayer $MoS_2$ with 75 atoms, bilayer graphene with 64 atoms and bilayer bismuthene with 36 atoms. For supercells with atom number larger than 1,000, only the Γ point was used. Meanwhile, a Monkhorst–Pack k-mesh of $1 \times 1 \times 29$ was used for CNTs and $MoS_2$ nanotubes, then $1 \times 1 \times 1$ ($2 \times 2 \times 1$) for TBGs (TBBs). SOC was considered in the calculation of bilayer bismuthene supercells and TBBs.

**Physical properties derived from the DFT Hamiltonian.** In a non-orthogonal atomic orbital basis set, the Hamiltonian and overlap matrix elements are defined as

$$H_{i\alpha,j\beta} = \langle \phi_{i\alpha} | \hat{H} | \phi_{j\beta} \rangle \tag{7}$$

and

$$S_{i\alpha,\,j\beta} = \langle \phi_{i\alpha} | \phi_{j\beta} \rangle, \qquad (8)$$

where $|\phi_{i\alpha}\rangle$ denotes the atomic orbital $\alpha$ of atom $i$. The DFT Hamiltonian matrix can be obtained from DFT self-consistent field calculations or predicted by the DeepH method. The overlap matrix is obtained by the inner product of the basis at very low computational cost. Accordingly, it is unnecessary to learn this quantity by neural network. After Fourier transformations of the Hamiltonian and overlap matrices, the eigenvalues $\mathcal{E}_{n\mathbf{k}}$ and eigenstates $v_{n\mathbf{k}}$ of the Hamiltonian $\hat{H}$ at band $n$ and wavevector $\mathbf{k}$ can be obtained by solving the generalized eigenvalue problem[33]

$$H(\mathbf{k})v_{n\mathbf{k}} = \mathcal{E}_{n\mathbf{k}} S(\mathbf{k})v_{n\mathbf{k}}. \qquad (9)$$

For moiré-twisted materials in the current study, the ARPACK library was used to compute a few eigenvalues of the large-scale sparse Hamiltonian matrix obtained from the DeepH method.

The 3D electric susceptibility $\chi$ and shift current conductivity $\sigma$[59] as functions of light frequency $\omega$ are given by

$$\chi^{ab} = \frac{e^2}{\epsilon_0 \hbar} \int \frac{d^3\mathbf{k}}{(2\pi)^3} \sum_{n,m} f_{nm} \frac{r_{nm}^a r_{mn}^b}{\omega_{mn}(\mathbf{k}) - \omega - i\eta} \qquad (10)$$

and

$$\sigma^{abc}(\omega) = \frac{\pi e^3}{\hbar^2} \int \frac{d^3\mathbf{k}}{(2\pi)^3} \times \sum_{n,m} f_{nm} \mathrm{Im}\left( r_{mn}^b r_{nm}^{c;a} + r_{mn}^c r_{nm}^{b;a} \right) \delta(\omega_{mn}(\mathbf{k}) - \omega), \quad (11)$$

where $a$, $b$ and $c$ are cartesian directions, $\epsilon_0$ is the vacuum permittivity, $\hbar$ is the reduced Planck's constant, $e$ is the charge of an electron and $\eta$ is an infinitesimal relaxation rate. $\omega_{nm}(\mathbf{k}) = \frac{E_{n\mathbf{k}} - E_{m\mathbf{k}}}{\hbar}$ and $f_{nm} = f_n(\mathbf{k}) - f_m(\mathbf{k})$ are the difference of energy eigenvalues and Fermi–Dirac occupations of bands $n$ and $m$ at wavevector $\mathbf{k}$, respectively. $r_{nm}^a$ and $r_{nm}^{b;a} = \frac{\partial r_{nm}^b}{\partial k^a} - i(r_{nn}^a - r_{mm}^a)r_{nm}^b$ are Berry connection and its general derivative, which are calculated with the DFT Hamiltonian using the method developed in ref. [33].

For low-dimensional systems, the response functions calculated by equations (10) and (11) need to be redefined to exclude the influence of the vacuum layer in the supercell. As we are interested in the susceptibility of 2D $MoS_2$ layers and quasi-1D CNTs and the shift current conductivity of 2D graphene layers, the 2D susceptibility, 1D susceptibility and 2D sheet conductivity are given by

$$\chi_{2D} = L_{sp} \times \chi_{3D}, \qquad (12)$$

$$\chi_{1D}^{\parallel} = S_{sp} \times \chi_{3D}^{\parallel}, \qquad (13)$$

and

$$\sigma_{2D} = L_{sp} \times \sigma_{3D}, \qquad (14)$$

respectively, where $S_{sp}$ and $L_{sp}$ are the cross-sectional area and height of the supercell and $\chi_{3d}^{\parallel}$ is the electric susceptibility along the periodic direction.

**Details on training the neural network.** Equations (3) and (4) include neural networks for updating the vertex and edge features. The neural network of the vertex is $\Phi_v^{(l)}(x) = \sigma(xW_1^{(l)} + b_1^{(l)}) \odot g(xW_2^{(l)} + b_2^{(l)})$, where the input is $x \in \mathbb{R}^{n_{in}}$, the weight is $W \in \mathbb{R}^{n_{in} \times n_{in}}$, the bias is $b \in \mathbb{R}^{n_{out}}$, $\odot$ denotes element-wise multiplication, $\sigma$ denotes the sigmoid function and $g$ denotes the softplus function[37]. The neural network of the edge is $\Phi_e^{(l)}(x) = \mathrm{SiLU}\left(xW_3^{(l)} + b_3^{(l)}\right)W_4^{(l)} + b_4^{(l)}$, which is a fully connected neural network including a hidden layer and a sigmoid linear unit (SiLU) activation function.

The MPNN model we use includes five MP layers, one LCMP layer and thus $471{,}409 + 129 \times N_{out}$ parameters, where $N_{out}$ is a hyperparameter of the number of selected orbital pairs. The cutoff radius $R_C$ for constructing crystal graphs is set to the cutoff radius of the corresponding atomic-like orbitals. The dimension of elemental embeddings, as well as vertex feature vectors in each layer, is set to 64. The initial edge features are a set of 128 Gaussian functions $\exp(-(|r_{ij}| - r_n)^2/\sigma^2)$, where the center $r_n$ is placed linearly between 0 and 6 Å, and $\sigma^2$ is set to 0.0044. The edge feature vector in each layer is a 128-dimensional vector. There are 25 real spherical harmonic functions $\{Y_{Jm}\}$ to expand orientation information, where $J$ is an integer ranging from 0 to 4, $m$ is an integer between $-J$ and $J$. Batch sizes of 12, 3, 4 and 1 are set for monolayer graphene, monolayer $MoS_2$, TBG and TBB, respectively. An Adam optimizer is used with a learning rate initiated at $1 \times 10^{-3}$, which later reduces to $2 \times 10^{-4}$ and finally to $4 \times 10^{-5}$. We implemented the MPNN model in DeepH method using the PyTorch-Geometric[60] Python library.

It is optional to learn $H'_{i\alpha,\,j\beta}$ separately or to treat $H'_{ij}$ as a whole. In the example study on monolayer graphene and TBG, multiple MPNN models were trained to represent the mapping from $\{\mathcal{R}\}_N$ to $H'_{ij}$ for different orbital pairs. For the $MoS_2$

and TBB, multi-dimensional vector outputs of a single MPNN model were used to represent Hamiltonian matrix blocks as a whole to achieve high efficiency.

## Data availability
Source data are provided with this Paper. The dataset used to train the deep-learning model is available at Zenodo[61].

## Code availability
The code used in the current study is available at GitHub (https://github.com/mzjb/DeepH-pack) and Zenodo[62].

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

## Acknowledgements

This work was supported by the Basic Science Center Project of NSFC (grant no. 51788104), the National Science Fund for Distinguished Young Scholars (grant no. 12025405), the National Natural Science Foundation of China (grant no. 11874035), the Ministry of Science and Technology of China (grant nos. 2018YFA0307100 and 2018YFA0305603), the Beijing Advanced Innovation Center for Future Chip (ICFC) and the Beijing Advanced Innovation Center for Materials Genome Engineering. M.Y. was supported by the Shuimu Tsinghua Scholar Program and Postdoctoral International Exchange Program. R.X. was funded by the China Postdoctoral Science Foundation (grant no. 2021TQ0187).

## Author contributions

Y.X. and W.D. proposed the project and supervised H.L., Z.W. and N.Z. in carrying out the research, with the help of M.Y., R.X. and X.G. All authors discussed the results. Y.X. and H.L. prepared the manuscript with input from the other co-authors.

## Competing interests

The authors declare no competing interests.

## Additional information

**Correspondence and requests for materials** should be addressed to Wenhui Duan or Yong Xu.

