## [Peer Review File · Nature Computational Science]

Peer Review Information

Journal: Nature Computational Science

Manuscript Title: Deep-Learning Density Functional Theory Hamiltonian for Efficient *ab initio* Electronic-Structure Calculation

Corresponding author name(s): Yong Xu, Wenhui Duan

Reviewer Comments & Decisions:

Decision Letter, initial version:

Date: 16th March 22 12:18:14

Last Sent: 16th March 22 12:18:14

Triggered By: Kaitlin McCardle

From: kaitlin.mccardle@us.nature.com

To: yongxu@mail.tsinghua.edu.cn

BCC: kaitlin.mccardle@us.nature.com

Subject: Decision on Nature Computational Science manuscript NATCOMPUTSCI-21-1103A-Z

Message: ** Please ensure you delete the link to your author homepage in this e-mail if you wish to forward it to your co-authors. **

Dear Professor Xu,

Your manuscript "Deep-Learning Density Functional Theory Hamiltonian for Efficient *ab initio* Electronic-Structure Calculation" has now been seen by 3 referees, whose comments are appended below. You will see that while they find your work of interest, they have raised points that need to be addressed before we can make a decision on publication.

The referees' reports seem to be quite clear. Naturally, we will need you to address all of the points raised.

While we ask you to address all of the points raised, the following points need to be substantially worked on:

- Please clarify the methodological novelty of your approach compared to previous

work

- Please perform additional quantitative comparisons against previous work as noted by Reviewers #1 and #3, and as discussed via email. For example, please include comparisons with Hegde and Bowen Sci. Rep. 7, 42669 (2017) (Noted by Reviewer #1) and other approaches such as Tensor-field networks, Cormorant, PhiSNet, spookynet etc (Noted by Reviewer #3)
- Please provide computational cost comparisons, as noted by Reviewer #3
- Please further clarify the generality of your method, and provide additional examples, if necessary.

Please use the following link to submit your revised manuscript and a point-by-point response to the referees' comments (which should be in a separate document to any cover letter):

[REDACTED]

** This url links to your confidential homepage and associated information about manuscripts you may have submitted or be reviewing for us. If you wish to forward this e-mail to co-authors, please delete this link to your homepage first. **

To aid in the review process, we would appreciate it if you could also provide a copy of your manuscript files that indicates your revisions by making use of Track Changes or similar mark-up tools. Please also ensure that all correspondence is marked with your Nature Computational Science reference number in the subject line.

In addition, please make sure to upload a Word Document or LaTeX version of your text, to assist us in the editorial stage.

To improve transparency in authorship, we request that all authors identified as 'corresponding author' on published papers create and link their Open Researcher and Contributor Identifier (ORCID) with their account on the Manuscript Tracking System (MTS), prior to acceptance. ORCID helps the scientific community achieve unambiguous attribution of all scholarly contributions. You can create and link your ORCID from the home page of the MTS by clicking on 'Modify my Springer Nature account'. For more information please visit www.springernature.com/orcid.

We hope to receive your revised paper within three weeks. If you cannot send it within this time, please let us know.

Best regards,

Kaitlin McCardle
Editor
Nature Computational Science

Reviewers comments:

Reviewer #1 (Remarks to the Author):

In this work, the authors have developed a message-passing neural network (MPNN) that predicts the DFT Hamiltonian orders of magnitude faster than DFT codes. The authors apply this methodology (DeepH) to graphene and a monolayer of MoSi₂. The authors obtain very accurate results for the band structures and electric susceptibility calculated from the predicted Hamiltonians for new atomic configurations not seen during training. Additionally, when the models are applied to some structures not seen during training, such as nanotubes, the accuracy is maintained. As a final test of the method, the authors study the case of twisted bilayer graphene and bismuthene, using only for training two-layer configurations displaced with respect to each other, but not twisted. The trained models are able to predict the Hamiltonian of bilayers with various twisting angles, with accuracy although in the case of the magic angle in twisted bilayer graphene, as new effects appear not observed during training, the accuracy is reduced, while still predicting some characteristic features.

Overall, I think the work is interesting and can be useful for the study of some electronic properties. However, I have some comments/questions about the work I would like the authors to address:

1) If I understood correctly, the message passing layers result in rotation invariant descriptors. The LCMP layer is then used to introduce orientation information based on bonds formed by the central atom to its neighbors. This approach of using the bonds between the central atom and its neighbors has been previously used in NN to introduce the orientation information for the prediction of other atomic properties such as in the work of Zhang et al. (Phys. Rev. Lett. 120, 143001 (2018)).

Additionally, the authors do not really comment on similar work of predicting DFT Hamiltonians with machine learning by Hedge and Bowen (ref 25). In that work, the DFT Hamiltonian is predicted for periodic structures of copper and diamond, obtaining very good results for the band structure. In that work, the treatment of the local environment in terms of neighbors is also used.

2) One concern I have is the large amount of data required to train each model for very specific cases spanning a very small area of configuration space (e.g. graphene sheet). Also, for each of those configurations obtained with VASP, the authors then have to run OpenMX to compute the DFT Hamiltonian using previously chosen atom-like basis functions. How expensive would this process be for large databases and more importantly, how flexible can DeepH be to learn the Hamiltonian for configurations spanning a large configuration space?

3) Following the last question, the work would greatly benefit if the authors showed the application of DeepH to 3D cases, to really demonstrate the generality of the method and its accuracy.

4) When studying the carbon nanotubes with the trained DeepH model, it would be interesting if the authors showed the PCA plot (or similar types of plots) of the descriptors from one of the final layers of the model as they represent the local structure of each atom and compare between those for graphene and those for the

nanotubes with different diameters. If the descriptors are very similar (especially in the large-diameter cases), then the transferability test is not as demanding. Otherwise, it is difficult to grasp how transferable is DeepH to other local atomic environments. The same can be said for the MoSi₂ case and the study of twisted van der Waals materials.

In all, I think the work is very interesting but it lacks examples showing the applicability to significantly different structures. This lack of examples may affect the impact of the work on the general materials' community.

Reviewer #2 (Remarks to the Author):

The paper describes a methodology for learning the Hamiltonian operator in the density functional theory using deep learning. My feeling is that there is really not very much new in this paper.

Moreover, I see a fundamental flaw here: Functionals in the density functional theory are supposed to be universal, yet only graphene is used as the training data. I think this is very limited. Universality is a very important part of modeling DFT. This is where the difference lies between a DFT level model and a PES level model.

Reviewer #3 (Remarks to the Author):

Deep-Learning Density Functional Theory
Hamiltonian for Efficient ab initio
Electronic-Structure Calculation - Review

In this work, the authors introduce a machine learning method for predicting the DFT Hamiltonian matrix for crystalline materials, bypassing the expensive self-consistent field iterations required by DFT calculations and thereby producing approximate DFT Hamiltonians at a significantly lower computational cost.

The paper is well written and the methods are explained in sufficient detail, especially with the support of the supplementary materials.

In order to deal with the large size and the rotational equivariance of the Hamiltonians, the authors use a message-passing neural network (MPNN) with localized environment representations, based off previous works dealing with the prediction of complex properties of atomistic systems. However, in this work the authors used the local environment representations in a clever and unique way in order to deal with specific challenges encountered in the particular scenario of crystalline materials.

Exploiting the near-sightedness of the Hamiltonian to constrain the size of the local atomic environments needed to accurately predict the entries of the Hamiltonian matrix is a clever idea, as are basis transformations of the predicted Hamiltonian block from a local to a global coordinate system in order to preserve equivariance and simplify learning. The basis transformation incurs some additional computational overhead, but avoids the need of having to use equivariant representations at all layers of the network, as is usually done in models of this nature.

The evaluations presented on various types of crystalline materials demonstrate the effectiveness of this method, showing that the properties derived from the predicted

DFT Hamiltonians closely match those obtained from actual DFT calculations. The experiments on twisted nanostructures are particularly interesting, showcasing the transferability of the model and implying that the model does in fact manage to learn some of the underlying physics of DFT and generalize to structures that are significantly different to those in the training set.

There are a few things that the authors could further elaborate or investigate in order to improve the paper.

- 1) The basis transformation of the local coordinate systems seems to be performed for every edge, which begs the question how much computational overhead is caused by the change of basis which involves Wigner matrix multiplications. The authors need to discuss whether this approach is more efficient and effective compared to approaches that propagate spherical harmonic features through the entire length of the network (such as Tensor-field networks, Cormorant, PhiSNet, spookynet etc) and show whether and why the approach used in this work is as expressive as those approaches.
- 2) Given that the authors include a subsection in the supplement focusing on the importance of the local coordinate message passing (LCMP) layer, it is a pity that they don't have specific ablation studies that show how much the model improves by introducing the LCMP layer. Such comparisons need to be included in order to underpin the importance of the final LCMP layer.

Author Rebuttal to Initial comments

Reviewer #1:

In this work, the authors have developed a message-passing neural network (MPNN) that predicts the DFT Hamiltonian orders of magnitude faster than DFT codes. The authors apply this methodology (DeepH) to graphene and a monolayer of MoSi₂. The authors obtain very accurate results for the band structures and electric susceptibility calculated from the predicted Hamiltonians for new atomic configurations not seen during training. Additionally, when the models are applied to some structures not seen during training, such as nanotubes, the accuracy is maintained. As a final test of the method, the authors study the case of twisted bilayer graphene and bismuthene, using only for training two-layer configurations displaced with respect to each other, but not twisted. The trained models are able to predict the Hamiltonian of bilayers with various twisting angles, with accuracy although in the case of the magic angle in twisted bilayer graphene, as new effects appear not observed during training, the accuracy is reduced, while still predicting some characteristic features.

Overall, I think the work is interesting and can be useful for the study of some electronic properties. However, I have some comments/questions about the work I would like the authors to address:

Response: We gratefully thank the referee for his/her careful review on our manuscript. We also thank the referee for pointing out that the work is interesting and for providing insightful and constructive comments that help us significantly improve the manuscript.

On the case study of magic-angle twisted bilayer graphene (TBG) (Fig. 6b), we would like to clarify that the appearance of observable differences between DFT (adapted from Ref. [42] using the plane-wave-basis code VASP) and DeepH (trained by the atomic-basis code OpenMX) is due to the inconsistency between different DFT codes. In fact, DeepH can well reproduce band structures of OpenMX for all our test calculations of TBGs with varying twist angles (Fig. 6b and Fig. S12).

In the following we will make a point-to-point response to the comments raised.

Comment 1: If I understood correctly, the message passing layers result in rotation invariant descriptors. The LCMP layer is then used to introduce orientation information based on bonds formed by the central atom to its neighbors. This approach of using the bonds between the central atom and its neighbors has been previously used in NN to introduce the orientation information for the prediction of other atomic properties such as in the work of Zhang et al. (Phys. Rev. Lett. 120, 143001 (2018)). Additionally, the authors do not really comment on similar work of predicting DFT Hamiltonians with machine learning by Hedge and Bowen (ref 25). In that work, the DFT Hamiltonian is predicted for periodic structures of copper and diamond, obtaining very good results for the band structure. In that work, the treatment of the local environment in terms of neighbors is also used.

Response: The referee is correct that the message passing layers result in rotation invariant descriptors and the local bonding orientation information is introduced into the LCMP layer. We thank the referee for pointing out a relevant reference about the incorporation of orientation information in neural network (NN), which we will discuss and cite explicitly in the revised manuscript. Deep learning methods based on NN usually use atomic bond lengths as rotation invariant descriptors to describe structure-property relationships. A few recent works have tried to introduce additional structural information into NN for improving the prediction of rotational-invariant physical properties, such as the total energy. As pointed out by the referee, the orientation information based on bonds between the central atom and its neighbors was introduced for the study of total energy by Zhang et al. [Ref. [9]: Phys. Rev. Lett. 120, 143001 (2018)]. In contrast, the quantity we considered here is covariant but not invariant under rotation (or gauge) transformations. The corresponding deep-learning task is more challenging, which would not work properly without an appropriate treatment of covariant requirements. We proposed to transform the DFT Hamiltonian matrix blocks to local coordinates and change the output labels of NN from rotation covariance to rotation invariance. Then one can apply transformation invariant NN to deal with transformation covariant quantities. As transformation invariance is a special case of transformation covariance, our approach generalizes the capability of NN to study a broader class of problems.

There are a few key differences between our work and that by Hegde and Bowen [Sci. Rep. 7, 42669 (2017)]. That work tried to learn the DFT Hamiltonian matrix by the kernel ridge regression (KRR) method. This statistic learning method has limited expressive power and is only applicable to simple material structures. More critically, that work does not provide an appropriate treatment of rotation covariance, which significantly limits the prediction accuracy. For instance, their approach works for simple structures involving *s* orbitals only (i.e., strained copper, without considering other orbitals), but cannot satisfactorily reproduce DFT band structures for materials involving other orbitals, like *p*-orbitals in the simple case study of unstrained diamond including two carbon atoms in the unit cell (Fig. R1). This is related to the fact that the DFT Hamiltonian matrix is invariant for *s* orbitals and gets covariant for other orbitals. In contrast, benefitting from the remarkable expressive power of deep neural network and an appropriate treatment of covariance requirements, our approach shows high prediction accuracy for much more

complex material structures including a large amount of atoms with irregular structure configurations and varying orbital functions.

Compare experiment: We performed new experiments to make quantitative comparisons with the referenced KRR method [Sci. Rep. 7, 42669 (2017)]. Specifically, we trained a DeepH model using similar dataset as the reference (i.e., 40 unit-cell diamond systems with random lattice strains within 4%), and used DeepH to make prediction on unstrained and strained diamond structures. DeepH shows high prediction accuracy for both DFT Hamiltonian matrix and band structure (Fig. R2). On the unstrained diamond, the averaged root mean square error (RMSE) of the predicted DFT Hamiltonian matrix elements is 0.6 meV, and the highest RMSE of p - p orbital coupling is 1.1 meV. In contrast, the p - p RMSE obtained by the KRR method is about 50 meV in the reference. Moreover, band structure of DFT can be well reproduced by DeepH, whereas the results of KRR show significant deviations in unoccupied bands (Fig. R1). It is thus concluded that DeepH outperforms the KRR method even for the study of simple material structures.

FIG. R1. Band structures of unstrained diamond computed by DFT and predicted by the kernel ridge regression (KRR) method (adapted from Ref. [Sci. Rep. 7, 42669 (2017)]). The results show significant deviations in unoccupied bands.

FIG. R2. Performance of DeepH on studying (a, b) unstrained and (c, d) 4% strained C-diamond. (a, c) Averaged root mean square error (RMSE) of $H'_{\alpha\beta}$ for different orbitals. (b, d) Band structures computed by DFT and DeepH.

As a response, we made the following changes:

- 1) One sentence was added in the last paragraph on page 4 to discuss Ref. [9] [Phys. Rev. Lett. 120, 143001 (2018)]: “*The orientation information based on bonds between the central atom and its neighbors was introduced for the study of total energy by Zhang et al. [9].*”
- 2) Two sentences were added in the last paragraph on page 8 to discuss Ref. [26] [Sci. Rep. 7, 42669 (2017)]: “*However, this method can hardly be applied to study more complex material systems due to the limited expressive power of statistical learning and the lack of an appropriate treatment of rotation covariance. A quantitative comparison between this method and DeepH is presented in the SI [34].*”
- 3) One paragraph and one figure (Fig. S21) were added in the Supplementary Information to make quantitative comparison with the referenced work (Ref. [26]).

Comment 2: One concern I have is the large amount of data required to train each model for very specific cases spanning a very small area of configuration space (e.g. graphene sheet). Also, for each of those configurations obtained with VASP, the authors then have to run OpenMX to compute the DFT Hamiltonian using previously chosen atom-like basis functions. How expensive would this process be for large databases and more importantly, how flexible can DeepH be to learn the Hamiltonian for configurations spanning a large configuration space?

Response: We thank the referee for the insightful comment. In fact, only a few hundred DFT calculations were employed for training in our example studies, since each DFT calculation provides a large amount of training data. Moreover, atomic configurations of DFT training calculations are not necessarily generated by *ab initio* molecular dynamics. They can also be obtained by randomly moving atoms away from equilibrium positions, and thus the computational cost for generating training structures can be saved. In this sense, it is not computationally expensive to generate the DFT training dataset.

On the comment about configuration space, deep neural network in principle can be applied to deal with complex problems with large configuration space due to its remarkable expressive power. The object of the present work is to learn the DFT Hamiltonian matrix as a function of atomic positions. For most

physical problems, only atomic configurations near equilibrium positions are concerned thermodynamically due to their relatively low energies. Thus we focused on configuration space near equilibrium for a given material. Solids with nearly periodic structures (like graphene with lattice vibrations) usually have small configuration space. The DeepH method can work well for such kind of systems: it could be applied for efficient electronic structure calculations of periodic or non-periodic geometries and could find valuable applications in studying lattice distortion effects, electron-lattice coupling, Moiré twist, etc. On the other hand, training a model in a large configuration space is indeed much more challenging, which usually requires more training data and possibly demands methodological improvement to achieve good accuracy.

New experiments: To test the performance of DeepH, we considered two kinds of material systems with a relatively larger configuration space:

1) Solid structures with multiple crystalline phases

We trained a DeepH model using a dataset containing different allotropes of carbon, including graphite and diamond (Fig. R3). To construct such dataset, 300 $3\times 3\times 3$ graphite supercells and 300 $3\times 3\times 3$ diamond supercells with random atomic displacements (up to 0.15 Å) with respect to equilibrium positions were prepared. A randomized training:validation:test split of 60:20:20 percent was used to test the quality of predictions. Here one unified neural network is applied to predict DFT Hamiltonian for two kinds of carbon allotropes. The mean absolute errors (MAEs) of H' for graphite and diamond are shown in Fig. R3(b) and Fig. R3(d), respectively. The MAEs averaged over all the orbital combinations for graphite and diamond are as low as 1.50 and 2.04 meV, respectively. The MAEs of DeepH do not increase with respect to that of graphene, though the configuration space gets larger.

2) Non-periodic molecular structures

Molecules contain many inequivalent atoms with distinct local chemical environment. Their configuration space considerably increases with increasing molecular size. We studied the DFT Hamiltonian problem for different molecules with strongly random structures obtained from *ab initio* molecular dynamics, including water (H_2O), ethanol (C_2H_6O), malondialdehyde ($C_3H_4O_2$), uracil ($C_4H_4N_2O_2$), and aspirin ($C_9H_8O_4$) (see details in the Supplementary Information). Their averaged MAEs of H' is 0.59, 0.60, 0.55, 0.47, and 0.83 meV, respectively. These MAEs are lower than that for graphene. These experiments suggest that DeepH is very likely applicable to the study of material systems spanning a large configuration space. We would like to do more critical experiments and developments in future works.

FIG. R3. Atomic structures (left panels) and mean absolute errors (MAEs) of $H'_{\alpha\beta}$ for different orbitals (right panels) of (a, b) graphite and (c, d) diamond.

As a response, we made the following changes:

- 1) Two paragraphs were added on page 9 to discuss the problem of large configuration space: “*Deep neural network in principle can be applied to deal with complex problems with large configuration space due to its remarkable expressive power...*”
- 2) Two paragraphs together with one figure (Fig. S16) and one table (Table S4) were added in the Supplementary Information to describe the new experiments.

Comment 3: Following the last question, the work would greatly benefit if the authors showed the application of DeepH to 3D cases, to really demonstrate the generality of the method and its accuracy.

Response: This is an instructive suggestion. In addition to 3D case studies about allotropes of carbon shown above, we also used DeepH to study 3D structures of bulk silicon. 300 $4\times 4\times 4$ bulk silicon supercells with random atomic displacements (up to 0.15 Å) with respect to equilibrium positions were prepared as dataset. We used the trained model to make prediction on a $5\times 5\times 5$ bulk silicon structure obtained by a 10 ps *ab initio* molecular dynamics (AIMD) simulation at 600 K with the time step of 1 fs. The averaged MAE of Hamiltonian matrix is 2.0 meV, and the band structure predicted by DeepH matches well with that by DFT (Fig. R4). Thus we conclude that DeepH is applicable to 3D material systems as well.

FIG. R4. Performance of DeepH on prediction for a $5\times 5\times 5$ bulk silicon supercell obtained by the AIMD simulation (atomic structure included in the Supplementary Information). (a) MAE of $H_{\alpha\beta}^i$ for different orbitals. (b) Band structure computed by DFT (red line) and DeepH (blue dot).

As a response, we made the following changes:

- 1) Two sentences were added in the first paragraph on page 9: “DeepH can also be applied to study material systems of other space dimensions. For instance, we have made experiments on 3D bulk materials (including silicon and allotropes of carbon) as well as quasi-0D molecules (see details in the SI [34]).”
- 2) One paragraph and one figure (Fig. S15) were added in the Supplementary Information to describe the new experiment on bulk silicon.

Comment 4: When studying the carbon nanotubes with the trained DeepH model, it would be interesting if the authors showed the PCA plot (or similar types of plots) of the descriptors from one of the final layers of the model as they represent the local structure of each atom and compare between those for graphene and those for the nanotubes with different diameters. If the descriptors are very similar (especially in the large diameter cases), then the transferability test is not as demanding. Otherwise, it is difficult to grasp how transferable is DeepH to other local atomic environments. The same can be said for the MoSi₂ case and the study of twisted van der Waals materials.

Response: This is a very helpful suggestion. As suggested by the referee, we performed principal component analysis (PCA) for the output atom features of the final message passing layer (i.e., input of the LCMP layer) on a random monolayer graphene supercell from training set and carbon nanotubes (CNTs) with different diameters (Fig. R5). 64-dimensional feature of each atom is mapped to one point in the PCA plot, which represents local structural information of atom. For the graphene supercell with randomly displaced geometry, there exist many kinds of local environments as indicated by the PCA plot. For CNTs, principal components (PCs) for different atoms in one CNT are localized at the same point due to structure symmetry. Here CNTs with larger diameters give larger PC1 value. Thus we guess that the value of PC1 to some extent reflects local curvature information of atomic structures. Remarkably, even for CNTs whose atomic features are far away from those of training data in the PCA plot, DeepH can well reproduce DFT calculation results for both Hamiltonian matrix elements (Fig. S5) and band structures (Fig. R5). This case study demonstrates the good generalization ability of DeepH, benefitting from the preservation of rotation covariance.

FIG. R5. PCA plot for atom features of the final LCMP layer, and the band structure comparisons by DFT and DeepH for CNT whose PCs are far away from those of random monolayer graphene.

Similar as for CNTs, we also performed PCA on one random monolayer MoS_2 from training set and a zigzag (50, 0) MoS_2 nanotube (Fig. R6). PCs for different atoms in the MoS_2 nanotube are localized at three points, which represent a Mo atom and two inequivalent S atoms. It appears that PC1 describes the element type, and PC2 describes the local structural stretching. Atomic features of MoS_2 nanotube are distinct from those of training data (monolayer MoS_2) in the PCA plot. The good prediction accuracy of DeepH on the MoS_2 nanotube demonstrates the generalization ability of DeepH.

FIG. R6. PCA plot for atom features of the final LCMP layer on one random monolayer MoS_2 supercell from training set and a zigzag (50, 0) MoS_2 nanotube.

PCA plot for atom features on random bilayer graphene from training set and twisted bilayer graphenes (TBGs) is displayed in Fig. R7(a), where no important diversity of atom features is shown for the two regular TBG systems. However, the situation changes if we perform PCA for bond features of the final LCMP layer, which is used to construct Hamiltonian matrix blocks in neural networks. Bonds features with atomic distances between 3.9 to 4.2 Å are localized in a small area (orange points in Fig. R7(b)), indicating that PCs for bond features mainly describe the interatomic distance. Furthermore, we

performed PCA for bond features focusing on the atomic distance interval between 3.9 to 4.2 Å. Then one may find that the PCs of bond features on TBGs are different from those of training data (Fig. R7(c)), illustrating the generalization ability of DeepH for studying varying bonding environments.

FIG. R7. PCA plot for (a) atom features of the final LCMP layer, (b) bond features of the final LCMP layer, and (c) bond features with atomic distances between 3.9 to 4.2 Å of the final LCMP layer on random bilayer graphene from training set ($\theta = 0^\circ$) and TBGs ($\theta \neq 0^\circ$).

PCA for atom features of the final LCMP layer on twisted bilayer bismuthenes (TBBs) is different from that on TBGs (Fig. R8). PCs on random bilayer bismuthene from training set or TBBs are localized in two small areas in the PCA plot, possibly corresponding to the surface and subsurface atoms. The PCs on the corresponding two areas for random bilayer bismuthene with zero twist angle (training set) are both far away from PCs for TBBs.

FIG. R8. PCA plot for atom features of the final LCMP layer on random bilayer bismuthene from training set and TBBs. The important diversity of atomic (and/or bond) features in PCA on nanotubes and twisted van der Waals materials and the good prediction accuracy on structures unseen in the training set demonstrate satisfactory generalization ability of DeepH.

As a response, we made the following changes:

- 1) One paragraph was added on page 9 to discuss PCA: “One may straightforwardly check the generalization ability of DeepH by performing principal component analysis (PCA) for the output atom features of the final MP layer or the output bond features of the final LCMP layer...”
- 2) A section including four paragraphs and four figures (Figs. S17-20) was added in the Supplementary Information to describe the PCA results.

In all, I think the work is very interesting but it lacks examples showing the applicability to significantly different structures. This lack of examples may affect the impact of the work on the general materials' community.

Response: We sincerely appreciate the referee for the insightful comments. As inspired by the referee, we performed a few new experiments. The promising results make us more optimistic about the future of DeepH. First, for the successful case studies (from 2D monolayer sheet to 1D nanotubes and from non-twisted to twisted bilayers) we learned by PCA that DeepH can make accurate predictions on new structures with PCs significantly different from the original training set. This demonstrated the generalization ability of DeepH more straightforwardly. Second, we further studied some more complex structures with relatively larger configuration spaces. The satisfactory results indicate that DeepH is a very flexible method and potentially has the ability to deal with complicated materials problems due to the remarkable expressive power of deep neural network. Third, as suggested by the referee, we generalized the application of DeepH to a few 3D systems, indicating that DeepH is applicable to different space dimensions. Last but not least, we proposed a theoretical framework to learn transformation covariant quantities via transformation invariant neural networks (NNs). Future development of DeepH would benefit from the great developments of transformation invariant NNs. Obviously the application of DeepH is not limited to the example case studies demonstrated in this preliminary work. In the final discussion part, we have suggested several important directions for future developments. We believe that the usefulness and enormous potential of DeepH would attract great interest from the general materials' community.

In a word, we have taken the suggestions of the reviewer and significantly revised the manuscript, in the hope that the reviewer may find the revised version satisfactory.

Reviewer #2:

The paper describes a methodology for learning the Hamiltonian operator in the density functional theory using deep learning. My feeling is that there is really not very much new in this paper.

Moreover, I see a fundamental flaw here: Functionals in the density functional theory are supposed to be universal, yet only graphene is used as the training data. I think this is very limited. Universality is a very important part of modeling DFT. This is where the difference lies between a DFT level model and a PES level model.

Response: We gratefully thank the referee for his/her critical comments. We believe that the present work makes important contribution to an emerging interdisciplinary direction, deep-learning *ab initio* electronic structure calculation. We developed a deep neural network approach to represent DFT Hamiltonian, so as to bypass the computationally demanding self-consistent field iterations of DFT and significantly improve the efficiency of *ab initio* electronic-structure calculation. Moreover, since we proposed a theoretical framework to learn transformation covariant quantities via transformation invariant neural networks (NNs), future

development of the method can benefit from the great developments of transformation invariant NNs. Our approach is general, useful and important. High accuracy, transferability and efficiency of the approach have been demonstrated for some example material systems. Our approach could be generally applied for efficient electronic-structure calculations of periodic or non-periodic geometries, which may find valuable applications in studying lattice distortion effects, electron-lattice coupling, Moiré twist, etc.

The capability of our approach is not limited to a specific example of graphene. We have carefully demonstrated the generality of the method by studying varying material systems, including flat or curved structures, non-twisted or twisted bilayers, and materials containing single or multiple elements, without or with strong spin-orbit coupling, etc. Moreover, we performed a few new experiments and demonstrated that DeepH is applicable to material systems of varying space dimensions, from quasi 0D (molecules), 1D (nanotubes), 2D (monolayer sheets) to 3D (bulk materials). With these diverse case studies, we hope the referee would find that the method is universal. Finally, the referee mentioned “a PES level model”. We guess that PES refers to “potential energy surface” by the referee. A potential energy surface describes the energy of a system, which seems irrelevant to the present research.

We hope the comments raised by the reviewer are properly addressed.

Reviewer #3:

In this work, the authors introduce a machine learning method for predicting the DFT Hamiltonian matrix for crystalline materials, bypassing the expensive self-consistent field iterations required by DFT calculations and thereby producing approximate DFT Hamiltonians at a significantly lower computational cost.

The paper is well written and the methods are explained in sufficient detail, especially with the support of the supplementary materials.

In order to deal with the large size and the rotational equivariance of the Hamiltonians, the authors use a message-passing neural network (MPNN) with localized environment representations, based off previous works dealing with the prediction of complex properties of atomistic systems. However, in this work the authors used the local environment representations in a clever and unique way in order to deal with specific challenges encountered in the particular scenario of crystalline materials.

Exploiting the near-sightedness of the Hamiltonian to constrain the size of the local atomic environments needed to accurately predict the entries of the Hamiltonian matrix is a clever idea, as are basis transformations of the predicted Hamiltonian block from a local to a global coordinate system in order to preserve equivariance and simplify learning. The basis transformation incurs some additional computational overhead, but avoids the need of having to use equivariant representations at all layers of the network, as is usually done in models of this nature.

The evaluations presented on various types of crystalline materials demonstrate the effectiveness of this method, showing that the properties derived from the predicted DFT Hamiltonians closely match those obtained from actual DFT calculations. The experiments on twisted nanostructures are particularly interesting, showcasing the transferability of the model and implying that the model does in fact manage to learn some of the underlying physics of DFT and generalize to structures that are significantly different to those in the training set.

There are a few things that the authors could further elaborate or investigate in order to improve the paper.

Response: We gratefully thank the referee for his/her careful review and expert summary on our manuscript. We also sincerely appreciate the referee for the high recognition and constructive comments on our work. Below we give a point-to-point response to the comments raised.

Comment 1: The basis transformation of the local coordinate systems seems to be performed for every edge, which begs the question how much computational overhead is caused by the change of basis which involves Wigner matrix multiplications. The authors need to discuss whether this approach is more efficient and effective compared to approaches that propagate spherical harmonic features through the entire length of the network (such as Tensor-field networks, Cormorant, PhiSNet, spookynet etc) and show whether and why the approach used in this work is as expressive as those approaches.

Response: We thank the referee for the insightful suggestion that helps us significantly improve the manuscript.

The basis transformation of Hamiltonian matrix blocks is to make simple linear algebra operations on small matrices. The transformation is done one time for a given material system and can be computed in parallel very efficiently. Thus, the computational cost for this process is insignificant compared with typical neural network computation. In contrast, the covariant neural networks based on spherical harmonic functions and group representation theory (such as Tensor-field networks, Cormorant, PhiSNet, etc.) require tensor products using Clebsch-Gordon coefficients in every layer of neural network during training and inference processes to ensure rotational covariance. The tensor-product computation could be very expensive, especially for large size systems and for calculations involving basis sets of high orbital angular momenta. As far as we know, applying these methods to study electronic structure of large-scale material systems remains elusive. Our method only needs to perform the basis transformation once before training process, which is computationally very efficient as discussed above. Moreover, benefitting from the rotation invariant nature of local coordinates, our approach can apply rotation invariant neural network to predict rotation covariant quantities, making the neural network architecture more flexible and efficient. Importantly, further development of the method would benefit from the great developments of transformation invariant neural network. Since all the important local bonding information, including bond length and orientation information, have been included as input, our method is expressive enough to achieve high prediction accuracy.

As mentioned by the referee, a few covariant neural networks have recently been developed, including Tensorfield networks, Cormorant, PhiSNet, spookynet, etc. Most of these works did not study the DFT Hamiltonian problem. After we posted our work on arXiv, the PhiSNet work appeared. PhiSNet applied a tensor-product based method to predict DFT Hamiltonian matrix for small molecules and is applicable to systems with fixed number of atoms. An earlier work (SchNOrb [Ref. 18]) studied the similar problem, which did not apply covariant neural network but used data augmentation to satisfy rotation covariance requirement.

New experiments: To make quantitative comparisons with SchNOrb and PhiSNet, we considered the same molecular structures obtained by *ab initio* molecular dynamics as in these two references, performed the electronic structure calculations by the OpenMX code, and used the dataset to train, validate and test DeepH. Five kinds of molecules were studied, including water (H₂O), ethanol (C₂H₆O), malondialdehyde (C₃H₄O₂), uracil (C₄H₄N₂O₂), and aspirin (C₉H₈O₄). The same number of training data were used as PhiSNet if not explicitly mentioned.

The comparisons of accuracy and efficiency are summarized in Table R1. Clearly, DeepH and PhiSNet have better performance than SchNOrb, due to the preservation of rotation covariance. The prediction MAE of DeepH and PhiSNet can be reduced to as low as sub meV, satisfactory for practical applications. According to the present tests, DeepH shows slightly lower (but comparable) accuracy due to the use of much smaller number of parameters than PhiSNet, whereas its efficiency is considerably higher (about one order of magnitude faster). Note that our neural network of DeepH was originally designed to study crystalline materials. Its computational parameters were not fully optimized for the study of molecular systems. Moreover, our theoretical framework can be applied to learn transformation covariant quantities via transformation invariant neural network. Further improvement of DeepH is possibly by borrowing ideas from more advanced rotation invariant neural network frameworks, such as SchNet, DimeNet, NequIP, PAINN, etc. We thus conclude that DeepH is as expressive as tensor-product-based neural networks but computationally more flexible and efficient.

Dataset	Inference time (s)			Number of parameters			Test set MAE (meV)		
	SchNorb	PhiSNet	DeepH	SchNorb	PhiSNet	DeepH	SchNorb	PhiSNet	DeepH
Water (H ₂ O, N_{train} : 500)	–	–	0.0005				4.501	0.479	1.048
Water (H ₂ O, N_{train} : 3000)	–	–	0.0005				–	–	0.593
Ethanol (C ₂ H ₆ O)	–	0.027	0.0023	~10 ⁷	~10 ⁷	~10 ⁵ -10 ⁶	5.099	0.331	0.601
Malondialdehyde (C ₃ H ₄ O ₂)	–	0.029	0.0022				5.200	0.335	0.547
Uracil (C ₄ H ₄ N ₂ O ₂)	–	0.050	0.0048				6.199	0.292	0.470
Aspirin (C ₉ H ₈ O ₄)	–	0.155	0.0148				13.77	0.349	0.833

TABLE R1. Comparison of performance with SchNOrb and PhiSNet for small molecule datasets.

As a response, we made the following changes:

- 1) Two paragraphs were added on page 9 to discuss PhiSNet: “*It is worth while comparing our method with covariant neural network methods (such as Tensor-field networks [27], Cormorant [28], PhiSNet [19], etc.), which are based on spherical harmonic functions and group representation theory...*”
- 2) A paragraph and a table (Table S4) were added in the Supplementary Information to make quantitative comparison with the referenced work (Ref. [18, 19]).
- 3) The works of Cormorant and SpookyNet were cited as Ref. [28] and Ref. [11], respectively.

Comment 2: Given that the authors include a subsection in the supplement focusing on the importance of the local coordinate message passing (LCMP) layer, it is a pity that they don't have specific ablation studies that show how much the model improves by introducing the LCMP layer. Such comparisons need to be included in order to underpin the importance of the final LCMP layer.

Response: This is a very valuable suggestion. As suggested by the referee, we performed ablation studies on the LCMP layer and found that the removal of LCMP layer would significantly decrease accuracy. Specifically, we trained two models for Mo-Mo orbital pairs of monolayer MoS₂ dataset, one with the original architecture and the other replacing the final LCMP layer by a normal message passing layer. Then we computed the mean squared errors (MSE) of Hamiltonian matrix for the test set. Our experiment indicates that the removal of LCMP layer increases MSE by about three orders of magnitudes, leading to much larger MAE (Table R2 and Fig. R9). Therefore, the introduction of LCMP layer into DeepH is critical to improving prediction accuracy.

	Test set MAE (meV)	Test set MSE (eV ²)
Baseline	0.8	0.000004
No LCMP layer	16.8	0.007931

TABLE R2. Results for ablation studies on the LCMP layer.

FIG. R9. MAEs of $H'_{\alpha, \beta}$ for different Mo-Mo orbital pairs of monolayer MoS₂ dataset using DeepH with (a) the original architecture and with (b) the LCMP layer replaced by one normal MP layer.

As a response, we made the following changes:

- 1) One sentence was added in the first paragraph on page 5: “*Note that the introduction of LCMP layer into DeepH is critical to improving prediction accuracy according to our test (supplementary Table S3).*”
- 2) A paragraph and a table (Table S3) were added in the Supplementary Information to describe the ablation studies of LCMP layer.

We have taken the suggestions of the reviewer and significantly revised the manuscript, in the hope that the reviewer may find the revised version satisfactory.

Summary of changes (Revisions are colored blue in the revised manuscript.)

- Two sentences were added in the last paragraph on page 8 to discuss Ref. [26] [Sci. Rep. 7, 426692017]: “*However, this method can hardly be applied to study more complex material systems due to the limited expressive power of statistical learning and the lack of an appropriate treatment of rotation covariance. A quantitative comparison between this method and DeepH is presented in the SI [34].*” One paragraph and one figure (Fig. S21) were added in the Supplementary Information to make quantitative comparison with the referenced work (Ref. [26]).
- Two paragraphs were added on page 9 to discuss PhiSNet: “*It is worth while comparing our method with covariant neural network methods (such as Tensor-field networks [27], Cormorant [28], PhiSNet [19], etc.), which are based on spherical harmonic functions and group representation theory...*” A paragraph and a table (Table S4) were added in the Supplementary Information to make quantitative comparison with the referenced work (Ref. [18, 19]).

- One sentence is added in the last paragraph on page 4 to discuss Ref. [9] [Phys. Rev. Lett. 120, 143001 (2018)]: “*The orientation information based on bonds between the central atom and its neighbors was introduced for the study of total energy by Zhang et al. [9].*”
- Two paragraphs were added on page 9 to discuss the problem of large configuration space: “*Deep neural network in principle can be applied to deal with complex problems with large configuration space due to its remarkable expressive power...*” One paragraphs together with one figure (Fig. S16) were added in the Supplementary Information to describe the new experiments of larger configuration space.
- Two sentences were added in the first paragraph on page 9: “*DeepH can also be applied to study material systems of other space dimensions. For instance, we have made experiments on 3D bulk materials (including silicon and allotropes of carbon) as well as quasi-0D molecules (see details in the SI [34]).*” A paragraph and a figure (Fig. S15) were added in the Supplementary Information to describe the new experiment on bulk silicon.
- One paragraph was added on page 9 to discuss PCA: “*One may straightforwardly check the generalization ability of DeepH by performing principal component analysis (PCA) for the output atom features of the final MP layer or the output bond features of the final LCMP layer...*” A section including four paragraphs and four figures (Figs. S17-20) was added in the Supplementary Information to describe the PCA results.
- The works of Cormorant and SpookyNet were cited as Ref. [28] and Ref. [11], respectively.
- One sentence is added in the first paragraph on page 5: “*Note that the introduction of LCMP layer into DeepH is critical to improving prediction accuracy according to our test (supplementary Table S3).*” A paragraph and a table (Table S3) were added in the Supplementary Information to describe the ablation studies of LCMP layer.

Decision Letter, first revision:

Date: 10th May 22 10:46:19

Last Sent: 10th May 22 10:46:19

Triggered By: Kaitlin McCardle

From: kaitlin.mccardle@us.nature.com

To: yongxu@mail.tsinghua.edu.cn

CC: computationalscience@nature.com

Subject: AIP Decision on Manuscript NATCOMPUTSCI-21-1103B

Message: Our ref: NATCOMPUTSCI-21-1103B

10th May 2022

Dear Dr. Xu,

Thank you for submitting your revised manuscript "Deep-Learning Density Functional Theory Hamiltonian for Efficient *ab initio* Electronic-Structure Calculation" (NATCOMPUTSCI-21-1103B). It has now been seen by the original referees and their comments are below. The reviewers find that the paper has improved in revision, and therefore we'll be happy in principle to publish it in Nature Computational Science,

pending minor revisions to satisfy the referees' final requests and to comply with our editorial and formatting guidelines.

TRANSPARENT PEER REVIEW

Nature Computational Science offers a transparent peer review option for new original research manuscripts submitted from 17th February 2021. We encourage increased transparency in peer review by publishing the reviewer comments, author rebuttal letters and editorial decision letters if the authors agree. Such peer review material is made available as a supplementary peer review file. **Please state in the cover letter 'I wish to participate in transparent peer review' if you want to opt in, or 'I do not wish to participate in transparent peer review' if you don't.** Failure to state your preference will result in delays in accepting your manuscript for publication. Please note: we allow redactions to authors' rebuttal and reviewer comments in the interest of confidentiality. If you are concerned about the release of confidential data, please let us know specifically what information you would like to have removed. Please note that we cannot incorporate redactions for any other reasons. Reviewer names will be published in the peer review files if the reviewer signed the comments to authors, or if reviewers explicitly agree to release their name. For more information, please refer to our [FAQ page](https://www.nature.com/documents/nr-transparent-peer-review.pdf).

Thank you again for your interest in Nature Computational Science Please do not hesitate to contact me if you have any questions.

Sincerely,

Kaitlin McCardle
Editor
Nature Computational Science

ORCID

Reviewer #1 (Remarks to the Author):

The authors have added new experiments and figures to the original publication which have increased the quality and impact of the work. Additionally, the authors have carefully addressed all the reviewers' questions. I believe that the revised manuscript merits publication.

Reviewer #2 (Remarks to the Author):

The paper has two ideas. The first is to use the local environment to deal with translation and rotational symmetry. The second is to train the Hamiltonian directly to avoid the expensive self-consistent iteration in density functional theory. If both ideas were new, I would recommend publication in Nature Computational Science. Unfortunately either is really new. The first, the idea of using the local environment to fix a local coordinate frame, has been used extensively before. One example that comes to mind is the first version of the so-called "Deep Potential":

http://www.global-sci.com/intro/article_detail/cicp/10541.html

The second idea has been used in the context of training tight binding Hamiltonian:

<https://arxiv.org/abs/2011.13774>

There are two difference with respect to the tight binding (TB) situation: 1) the TB Hamiltonian is acting on the localized Wannier function, while DeepH is acting on a predefined basis.

2) the TB Hamiltonian has to be trained on the energy spectra, while DeepH can be trained directly by the entry of the Hamiltonian matrix.

There are advantages and disadvantages for these two designs. The main concern for DeepH is that it is limited to a given basis, which is not the most common plan wave, and compare to TB it is more expensive. While the advantage is that it can use more training label and the training should be more stable. As a result, they could do more complicated experiments on larger systems.

Overall I think this is a solid piece of work, and it would be a nice contribution to something like JCTC. But it is not at the level of Nature Computational Science.

Reviewer #3 (Remarks to the Author):

In the revised version, the authors have adequately addressed all the remarks with regards to the comparison to other equivariant models and ablation studies about the local coordinate message passing (LCMP) layer.

For the first remark, the authors compared the accuracy of their model to two neural networks designed for predicting molecular Hamiltonians, showing that their model provides a good computational efficiency vs. accuracy tradeoff, where the accuracy is much better than the non-equivariant SchNOrb and achieves accuracy levels close to the equivariant model PhiSNet. While the accuracy is somewhat lower compared to the equivariant neural network in all examples, the computational cost of the proposed model is lower since it bypasses the need for equivariant tensor-products using the Clebch-Gordan coefficients.

Additionally, the authors also demonstrate that the final LCMP layer that re-introduces some information about the orientation of the molecule is crucial to their network, showing that the accuracy of their model drops by at least an order of magnitude with the replacement of this layer with a regular message passing layer.

Furthermore, the authors have included a series of further analyses and experiments as a response to other reviewers which demonstrate a fairly wide applicability for their model.

Given these changes in the revised version, I would advise to accept the ms.

Author Rebuttal, first revision:

Reviewer #1:

The authors have added new experiments and figures to the original publication which have increased the quality and impact of the work. Additionally, the authors have carefully addressed all the reviewers' questions. I believe that the revised manuscript merits publication.

Response: We gratefully thank the referee for his/her supportive comments.

Reviewer #2:

The paper has two ideas. The first is to use the local environment to deal with translation and rotational symmetry. The second is to train the Hamiltonian directly to avoid the expensive self-consistent iteration in density functional theory. If both ideas were new, I would recommend publication in Nature Computational Science. Unfortunately either is really new.

Response: We thank the referee for his/her review on our manuscript. In the following we will make a point-to-point response to the comments raised.

The first, the idea of using the local environment to fix a local coordinate frame, has been used extensively before. One example that comes to mind is the first version of the so-called "Deep Potential": http://www.global-sci.com/intro/article_detail/cicp/10541.html

Response: The idea of using information of local chemical environment is not new for the learning of gauge invariant quantities, as total energy in the work of "Deep Potential". This fact was clearly mentioned in our manuscript. However, rare work has been tried to apply deep neural networks to learning rotation covariant (not invariant) quantity, such as the DFT Hamiltonian matrix. An invention of our work is to propose a flexible and efficient theoretical framework that applies rotation invariant neural network to learn rotation covariant quantities via local coordinate transformations. This novelty was appreciated by the other referees.

The second idea has been used in the context of training tight binding Hamiltonian: <https://arxiv.org/abs/2011.13774>

There are two difference with respect to the tight binding (TB) situation: 1) the TB Hamiltonian is acting on the localized Wannier function, while DeepH is acting on a predefined basis. 2) the TB Hamiltonian has to be trained on the energy spectra, while DeepH can be trained directly by the entry of the Hamiltonian matrix.

Response: There are crucial differences between our work and the reference (arXiv: 2011.13774):

1) The referenced work did not learn the tight binding (TB) Hamiltonian and instead only trained the eigenvalues of TB Hamiltonian (i.e., energy spectra), because the authors believed that the latter quantities are gauge invariant and thus easier to learn. By this strategy the gauge covariance problem of Hamiltonian matrix was circumvented. However, because the nearsightedness principle does not apply to energy spectra, the generalization ability of learning energy spectra would be seriously limited. It is difficult, if not impossible, to generalize the method to complex material systems. In contrast, we provided a theoretical framework to deal with rotation covariance requirements of DFT Hamiltonian matrix. Importantly, our method could benefit from the nearsighted nature of electronic matter, thus ensuring high prediction accuracy and good transferability.

2) Despite the above key difference, it should be noted that rotation transformation of Wannier basis functions is not well defined, which is disadvantageous for the training of DFT Hamiltonian matrix. Moreover, heavy human intervention is usually required to obtained Wannier-based TB Hamiltonian, which could further complicate the computation. In contrast, our method does not suffer from such kind of disadvantages.

There are advantages and disadvantages for these two designs. The main concern for DeepH is that it is limited to a given basis, which is not the most common plane wave, and compare to TB it is more expensive. While the advantage is that it can use more training label and the training should be more stable. As a result, they could do more complicated experiments on larger systems. Overall I think this is a solid piece of work, and it would be a nice contribution to something like JCTC. But it is not at the level of Nature Computational Science.

Response: On the disadvantages mentioned by the referee, we would like to clarify a few points:

1) DeepH is aimed to reproduce *ab initio* tight-binding (TB) Hamiltonian of DFT. Thus, there is no need to fit tight-binding Hamiltonian for training. In contrast, the fitting of Wannier-based TB Hamiltonian usually demands heavy human intervention, which makes the training calculations much more complicated.

2) It should be noted that DFT codes using localized basis have been widely applied in the community, as popular as those using plane wave basis. Thus, we do not consider using DFT codes with localized basis as a disadvantage of DeepH.

3) In Supplementary Table 5, we compared the performance of DeepH with other methods. The computation of DeepH, though a little bit more expensive than traditional TB methods, remains very cheap and efficient due to the relatively small size of basis set and the sparse nature of TB Hamiltonian matrix.

On the other hand, we thank the referee for pointing out, as an advantage of our DeepH method, that DeepH can use more training label and the training process should be more stable. DeepH takes full advantage of the nearsightedness principle of electronic matter and can properly deal with rotation covariance of DFT Hamiltonian matrix, enabling the prediction of larger or more complex systems than training data.

We hope the comments raised by the reviewer are properly addressed.

Reviewer #3:

In the revised version, the authors have adequately addressed all the remarks with regards to the comparison to other equivariant models and ablation studies about the local coordinate message passing (LCMP) layer.

For the first remark, the authors compared the accuracy of their model to two neural networks designed for predicting molecular Hamiltonians, showing that their model provides a good computational efficiency vs. accuracy tradeoff, where the accuracy is much better than the non-equivariant SchNOrb and achieves accuracy levels close to the equivariant model PhiSNet. While the accuracy is somewhat lower compared to the equivariant neural network in all examples, the computational cost of the proposed model is lower since it bypasses the need for equivariant tensor-products using the Clebch-Gordan coefficients. Additionally, the authors also demonstrate that the final LCMP layer that re-introduces some information about the orientation of the molecule is crucial to their network, showing that the accuracy of their model drops by at least an order of magnitude with the replacement of this layer with a regular message passing layer.

Furthermore, the authors have included a series of further analyses and experiments as a response to other reviewers which demonstrate a fairly wide applicability for their model.

Given these changes in the revised version, I would advise to accept the ms.

Response: We gratefully thank the referee for his/her supportive comments.

Final Decision Letter:

Date: 17th May 22 11:57:03

Last Sent: 17th May 22 11:57:03

Triggered By: Kaitlin McCardle

From: kaitlin.mccardle@us.nature.com

To: yongxu@mail.tsinghua.edu.cn

Subject: Decision on Nature Computational Science manuscript NATCOMPUTSCI-21-1103C

Message: Dear Professor Xu,

We are pleased to inform you that your Article "Deep-Learning Density Functional Theory Hamiltonian for Efficient *ab initio* Electronic-Structure Calculation" has now been accepted for publication in Nature Computational Science.

Please note that we have made modifications to the text to remove overhyped words, such as "remarkable"/"remarkably" and "extremely well." I have attached the modified text, for your reference.

Please note that *Nature Computational Science* is a Transformative Journal (TJ). Authors may publish their research with us through the traditional subscription access route or make their paper immediately open access through payment of an article-processing charge (APC). Authors will not be required to make a final decision about access to their article until it has been accepted. [Find out more about Transformative Journals](https://www.springernature.com/gp/open-research/transformative-journals)

Authors may need to take specific actions to achieve [compliance with funder and institutional open access mandates](https://www.springernature.com/gp/open-research/funding/policy-compliance-faqs). If your research is supported by a funder that requires immediate open access (e.g. according to [Plan S principles](https://www.springernature.com/gp/open-research/plan-s-compliance)) then you should select the gold OA route, and we will direct you to the compliant route where possible. For authors selecting the subscription publication route, the journal's standard licensing terms will need to be accepted, including [self-archiving policies](https://www.springernature.com/gp/open-research/policies/journal-policies). Those licensing terms will supersede any other terms that the author or any third party may assert apply to any version of the manuscript.

Acceptance of your manuscript is conditional on all authors' agreement with our publication policies (see <https://www.nature.com/natcomputsci/for-authors>). In particular your manuscript must not be published elsewhere and there must be no announcement of the work to any media outlet until the publication date (the day on which it is uploaded onto our web site).

Before your manuscript is typeset, we will edit the text to ensure it is intelligible to our wide readership and conforms to house style. We look particularly carefully at the titles of all papers to ensure that they are relatively brief and understandable.

Once your manuscript is typeset and you have completed the appropriate grant of rights, you will receive a link to your electronic proof via email with a request to make any corrections within 48 hours. If, when you receive your proof, you cannot meet this deadline, please inform us at rjsproduction@springernature.com immediately.

If you have queries at any point during the production process then please contact the production team at rjsproduction@springernature.com. Once your paper has been scheduled for online publication, the Nature press office will be in touch to confirm the details.

Content is published online weekly on Mondays and Thursdays, and the embargo is set at 16:00 London time (GMT)/11:00 am US Eastern time (EST) on the day of publication. If you need to know the exact publication date or when the news embargo will be lifted, please contact our press office after you have submitted your proof corrections. Now is the time to inform your Public Relations or Press Office about your paper, as they might be interested in promoting its publication. This will allow them time to prepare an accurate and satisfactory press release. Include your manuscript tracking number NATCOMPUTSCI-21-1103C and the name of the journal, which they will need when they contact our office.

About one week before your paper is published online, we shall be distributing a press release to news organizations worldwide, which may include details of your work. We are happy for your institution or funding agency to prepare its own press release, but it must mention the embargo date and Nature Computational Science. Our Press Office will contact you closer to the time of publication, but if you or your Press Office have any inquiries in the meantime, please contact press@nature.com.

We welcome the submission of potential cover material (including a short caption of around 40 words) related to your manuscript; suggestions should be sent to Nature Computational Science as electronic files (the image should be 300 dpi at 210 x 297 mm in either TIFF or JPEG format). We also welcome suggestions for the Hero Image, which appears at the top of our [home page](http://www.nature.com/natcomputsci); these should be 72 dpi at 1400 x 400 pixels in JPEG format. Please note that such pictures should be selected more for their aesthetic appeal than for their scientific content, and that colour images work better than black and white or grayscale images. Please do not try to design a cover with the Nature Computational Science logo etc., and please do not submit composites of images related to your work. I am sure you will understand that we cannot make any promise as to whether any of your suggestions might be selected for the cover of the journal.

Best regards,

Kaitlin McCardle
Editor
Nature Computational Science

P.S. Click on the following link if you would like to recommend Nature Computational Science to your librarian: https://www.springernature.com/gp/librarians/recommend-to-your-library

** Visit the Springer Nature Editorial and Publishing website at www.springernature.com/editorial-and-publishing-jobs for more information about our career opportunities. If you have any questions please click here. **